# ENHANCING MULTI-IMAGE UNDERSTANDING THROUGH DELIMITER TOKEN SCALING

**Minyoung Lee[1], Yeji Park[1], Dongjun Hwang[1], Yejin Kim[1,2], Seong Joon Oh[2,3], Junsuk Choe[1†]**
[1]Sogang University, [2]KAIST, [3]University of Tübingen

## ABSTRACT

Large Vision-Language Models (LVLMs) achieve strong performance on single-image tasks, but their performance declines when multiple images are provided as input. One major reason is the *cross-image information leakage*, where the model struggles to distinguish information across different images. Existing LVLMs already employ delimiter tokens to mark the start and end of each image, yet our analysis reveals that these tokens fail to effectively block cross-image information leakage. To enhance their effectiveness, we propose a method that scales the hidden states of delimiter tokens. This enhances the model's ability to preserve image-specific information by reinforcing intra-image interaction and limiting undesired cross-image interactions. Consequently, the model is better able to distinguish between images and reason over them more accurately. Experiments show performance gains on multi-image benchmarks such as Mantis, MuirBench, MIRB and QBench2. We further evaluate our method on text-only tasks that require clear distinction. The method improves performance on multi-document and multi-table understanding benchmarks, including TQABench, MultiNews and WCEP-10. Notably, our method requires no additional training or inference cost. Code is available at: https://github.com/MYMY-young/DelimScaling

## 1 INTRODUCTION

Large Vision-Language Models (LVLMs) demonstrate strong image understanding capabilities when a single image is provided (Li et al., 2023a; Liu et al., 2023). However, their performance significantly degrades when multiple images are given as input (Zeng et al., 2025; Jiang et al., 2024). A recent study (Park et al., 2025) attributes this degradation to the model's inability to clearly distinguish between individual images, a phenomenon referred to as *cross-image information leakage*. As a result, the generated output often intermixes information across different images.

While existing models introduce special *image delimiter tokens* to separate images, the role and mechanism of these tokens remain largely unexplored in the literature. To address this gap, we analyze how delimiter tokens function within the model. Our analysis of attention scores shows that although these tokens help distinguish images to some extent, cross-image interaction persists. This indicates that current models struggle to fully isolate visual contexts across images, ultimately leading to information leakage.

To better understand this behavior, we examine how delimiter tokens contribute to image separation and identify two key properties: their ability to absorb attention from other image tokens and their role in reinforcing intra-image interaction. Based on these insights, we propose a simple yet effective method that strengthens both properties by scaling the hidden states of delimiter tokens. This approach reduces cross-image interaction while preserving intra-image interaction, thereby helping the model distinguish between images more effectively.

To validate our findings, we apply the proposed method to a range of multi-image understanding tasks. Our approach significantly improves performance on benchmark datasets such as Mantis (Jiang et al., 2024), MuirBench (Wang et al., 2024a), MIRB (Zhao et al., 2024), and QBench2 (Zhang et al., 2024). Furthermore, we observe consistent gains in text-only scenarios

---

[†]Corresponding author.

where clear separation is also essential, such as multi-table and multi-document tasks. The method improves accuracy on TQABench (Qiu et al., 2024), MultiNews (Fabbri et al., 2019), and WCEP-10 (Ghalandari et al., 2020). Notably, these improvements are achieved without any additional training or inference overhead, highlighting the practicality and efficiency of our approach.

## 2 RELATED WORK

### 2.1 MULTI-IMAGE UNDERSTANDING

Recently, there has been active research on multi-image understanding in Large Vision-Language Models (LVLMs). One line of work focuses on training-based approaches to improve performance on multi-image tasks. For example, Jiang et al. (2024) constructed a multi-image instruction dataset to address the performance gap between single-image and multi-image understanding, and achieved performance gains through supervised fine-tuning. However, such training-based approaches face two major limitations: the high cost of curating high-quality multi-image instruction data and the need for substantial computational resources.

To overcome these limitations, training-free approaches have also been proposed. AVAM (Zeng et al., 2025) points out that visual redundancy becomes more severe in multi-image settings. It mitigates this issue by using text–image alignment to select only the most relevant visual regions. However, its reliance on an external text encoder and separate preprocessing module introduces structural complexity and limits flexibility. FOCUS (Park et al., 2025), another training-free method, tackles confusion between images by introducing a contrastive decoding strategy that separates outputs based on image-specific contexts. While effective, this approach requires $n + 1$ forward passes for $n$ images, leading to high inference costs.

In contrast, our method aims to enhance multi-image understanding without requiring additional training, inference-time overhead, or architectural modifications.

### 2.2 SINK TOKENS IN LARGE LANGUAGE MODELS

Recent studies on large language models have drawn attention to the phenomenon where certain tokens exhibit unusually high activations (Gu et al., 2024; Guo et al., 2024; Sun et al., 2024; Barbero et al., 2025). These *sink tokens*, often placed at the beginning of input sequences, tend to receive strong activations that act as implicit bias terms—uniformly influencing attention patterns across the sequence (Sun et al., 2024).

Although research on token-level irregular behaviors has been active, most prior work has focused on the `<BOS>` sink token in text-based LLMs. In contrast, our paper presents new observations and detailed characteristics of Image Delimiter Tokens, which have not been analyzed previously, offering a perspective that differs from existing sink-token studies.

We examine how LVLMs behave differently from LLMs when processing multi-image inputs. Similar to LLMs, the first token in LVLMs is also a sink token and receives high attention. (Kang et al., 2025) However, due to the structural properties of multi-image inputs, the sink patterns observed in LLMs do not fully generalize to LVLMs. The delimiter tokens inserted to separate multiple images each receive substantial attention, and as a result, the relative amount of attention allocated to the conventional sink token decreases compared to the case without delimiter tokens.

Furthermore, although delimiter tokens resemble sink tokens in that they receive high attention, their operational behavior is distinct. Unlike conventional sink tokens that influence the entire input globally, delimiter tokens attend primarily to tokens within their corresponding image, functioning as localized bias terms. This sink-like but localized behavior is unique to LVLMs under multi-image settings and has not been explored in prior work, largely because earlier studies focused on text-only or single-image inputs. Building on these observations, our paper provides an in-depth analysis of how delimiter tokens shape and regulate the attention structure for each image in multi-image prompts.

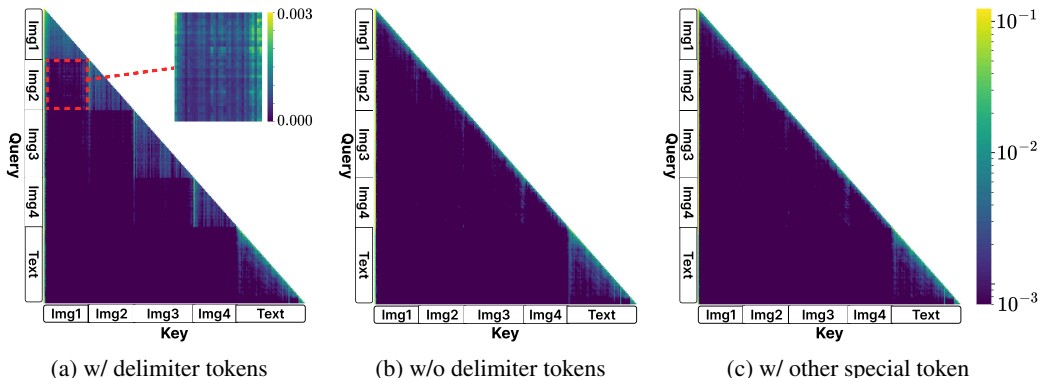

Figure 1: Impact of image delimiter tokens on attention maps. (a) With delimiter tokens, clear triangular patterns mark image boundaries. (b) Without them, these patterns disappear. (c) Replacing them with other special tokens (`<|im_start|>`) yields the same effect.

## 2.3 CROSS-IMAGE INFORMATION LEAKAGE

*Cross-image information leakage* refers to the phenomenon where a model fails to clearly separate multiple input images, resulting in unintended mixing of information across them. This issue was first reported by Park et al. (2025), which confirmed its existence but did not analyze the underlying cause. In this work, we examine the attention patterns of image delimiter tokens and offer a detailed analysis of how cross-image information leakage arises inside the model. Our findings reveal how delimiter tokens behave during multi-image processing and motivate the design of an effective strategy to mitigate cross-image information leakage.

## 3 DO IMAGE DELIMITER TOKENS REALLY WORK?

Cross-image information leakage exists despite the use of special tokens designed to distinguish individual images (e.g., `<|vision_start|>` and `<|vision_end|>` in Qwen2.5-VL). To investigate the cause of this phenomenon, we analyze the behavior of these *image delimiter tokens*. Our goal is to determine whether these tokens enable the model to discriminate between images and to identify their limitations.

**Do Image Delimiter Tokens Function as Intended to Distinguish Images?**  To assess whether image delimiter tokens fulfill their intended role, we remove them and observe the resulting changes in the attention score map. This allows us to assess how the presence or absence of these tokens affects attention patterns across multiple images.

As shown in Figure 1a, when delimiter tokens are present, the attention map exhibits distinct triangular block patterns that clearly delineate image boundaries. In contrast, removing the delimiter tokens (Figure 1b) eliminates these triangular patterns, making it difficult to distinguish which tokens belong to which image. Similar results are observed when the image delimiter tokens are replaced with other special tokens commonly used in LVLMs. For example, in Figure 1c, replacing `<|vision_start|>` and `<|vision_end|>` with `<|im_start|>`, a token typically used to indicate the start of a message, results in the disappearance of triangular patterns and image-wise confusion. Additional experiments using other special tokens exhibit the same pattern, with full details provided in Appendix A.1.

We also find that the presence of triangular attention patterns correlates with model performance: removing or replacing the image delimiter tokens leads to a performance drop of approximately 10 percentage points, as shown in Appendix A.1. These findings suggest that image delimiter tokens play a critical role in enabling the model to distinguish images in multi-image LVLMs. Without them, the model fails to form clear attention boundaries and exhibits notable performance degradation, underscoring their importance.

**Limitations of Image Delimiter Tokens.** While the preceding analysis confirms that image delimiter tokens support distinguishing between images—both in attention maps and performance—they do not fully prevent interactions across different images. Specifically, we observe some degree of cross-image interaction in the attention score map (see the red box in Figure 1a). This suggests that although the tokens help distinguish between images, their distinguishing effect remains incomplete. To address this limitation, we conduct a more detailed analysis of their behavior and identify two key properties that guide the design of our method.

## 4 IMAGE-WISE TAGGING VIA DELIMITER TOKENS

We analyze the attention patterns of image delimiter tokens in multi-image LVLM inputs and uncover two key properties that contribute to distinguishing images.

> **Property 1:** The $i$-th image delimiter token receives strong attention from the tokens of the $i$-th image, forming a correspondence between the delimiter token and the image.

As shown in Figure 1a, tokens in the $i$-th image consistently attend to the $i$-th delimiter token, forming a distinct vertical stripe in the attention map. This localized attention pattern contrasts with the global behavior of sink tokens, which attract attention from all tokens. Instead, each delimiter token is predominantly attended to by tokens from a single image, establishing a clear one-to-one mapping between images and delimiter tokens. Figure 2a further confirms this: the $i$-th image delimiter token receives strong attention from its corresponding image, while receiving little attention from others.

> **Property 2:** The strong attention of image delimiter tokens serves as an *image tag*, thereby reinforcing intra-image interaction.

Based on Property 1, we interpret each delimiter token as an image-specific tag associated with a particular image block. This tagging effect strengthens intra-image interaction. As shown in Figure 1a, the triangular patterns in the attention map reflect intensified mutual attention among tokens within the same image. In contrast, Figures 1b and 1c—where the delimiter tokens are removed or replaced—show weaker, more diagonal-dominant structures, indicating reduced intra-image interaction.

The mechanism behind this is illustrated in Equation 1, where the attention output is expressed as the weighted sum of value vectors:

$$\text{Attention}(Q_q, K_{\leq q}, V_{\leq q}) = \sum_{i \leq q} p_{q,i}\, v_i = \sum_{d \leq q} p_{q,d}\, v_d + \sum_{j \leq q} p_{q,j}\, v_j, \quad d \in \mathcal{D},\ j \notin \mathcal{D}. \quad (1)$$

In this formulation, $q$ and $i$ denote token indices. The term $p_{q,i}$ represents the attention score assigned by query $q$ to token $i$. The set $\mathcal{D} = \{d_1, d_2, \ldots\}$ contains the indices of delimiter tokens. For notational simplicity, we omit the query index $q$ in the following explanation.

Due to the attention pattern established in Property 1, each token in image $i$ assigns a dominant weight to its associated delimiter $d_i$. As a result, all tokens within an image share a common additive term $p_{d_i} v_{d_i}$ in the attention output, effectively functioning as a localized bias that enhances intra-image interaction. The rightmost plot of Figure 2b shows that for Image 3, $p_{d_3} v_{d_3}$ is about 15 times and 30 times larger than $p_{d_2} v_{d_2}$ and $p_{d_1} v_{d_1}$, respectively, confirming its dominant effect on the output.

We refer to this mechanism as *image-wise tagging*: each delimiter token contributes a localized,

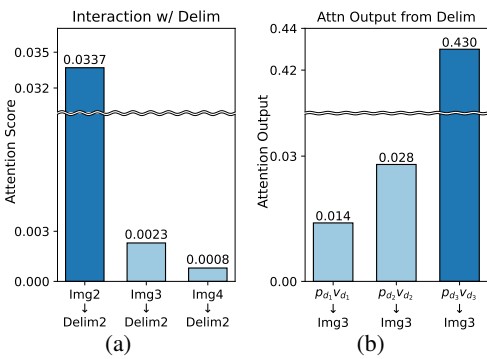

Figure 2: (a) Attention to the second-image delimiter. (b) Image tagging values.

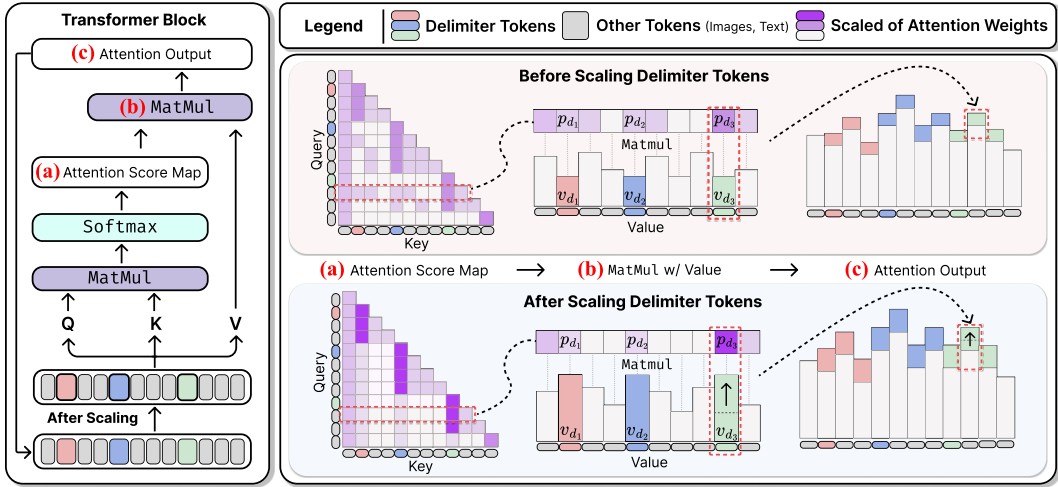

Figure 3: Effect of scaling image delimiter tokens on attention. Left: attention computation flow in a transformer block. Right: before scaling (top), delimiter tokens receive limited attention, leading to cross-image leakage. After scaling (bottom), delimiter tokens act as strong attractors (like sink tokens), distinguishing between images while preserving intra-image interactions (Property 2).

shared bias to the attention outputs of its associated image. This common additive term is shared across all tokens within the same image, thereby reinforcing intra-image interaction. As shown in the before-scaling part of Figure 3c, this shared term can be clearly observed being added to the attention outputs.

## 5 METHOD

Based on the two key properties of image delimiter tokens, we introduce a simple yet effective method that strengthens the model's ability to discriminate between images without compromising intra-image interactions. Specifically, we explore a simple strategy to reinforce the importance of delimiter tokens during attention computation. Among various possible implementations, we adopt a particularly simple approach: scaling the hidden states of delimiter tokens.

Let $h_t^{(l)} \in \mathbb{R}^d$ denote the hidden state of token $t$ at layer $l$, and let $\mathcal{D}$ denote the set of image delimiter token indices. We modify the hidden states as follows ($\lambda > 1$ is a scaling factor):

$$h_t^{(l)*} = \begin{cases} \lambda \cdot h_t^{(l)} & \text{if } t \in \mathcal{D}, \\ h_t^{(l)} & \text{otherwise.} \end{cases} \tag{2}$$

Our hidden state scaling method amplifies both properties of delimiter tokens discussed in Section 4. We adopt this approach due to its simplicity, strong empirical impact, and compatibility with standard attention implementations such as FlashAttention (Dao, 2023). Other variants such as layer-specific scaling, scaling query or value vectors directly, and modifying attention scores are also feasible. However, directly modifying the attention scores requires substantial resources, as discussed in Section 5.3. In comparison, we find that hidden state scaling strikes a favorable balance between effectiveness and efficiency. We consider these alternatives as promising directions for future research. Below, we analyze how this mechanism enhances the role of delimiter tokens in multi-image attention.

### 5.1 HOW OUR METHOD ENHANCES DELIMITER TOKEN PROPERTIES

Scaling the hidden states of image delimiter tokens reinforces Property 1 by increasing the attention they receive. This follows the general principle behind sink tokens (Gu et al., 2024), where higher activation leads to stronger attention.

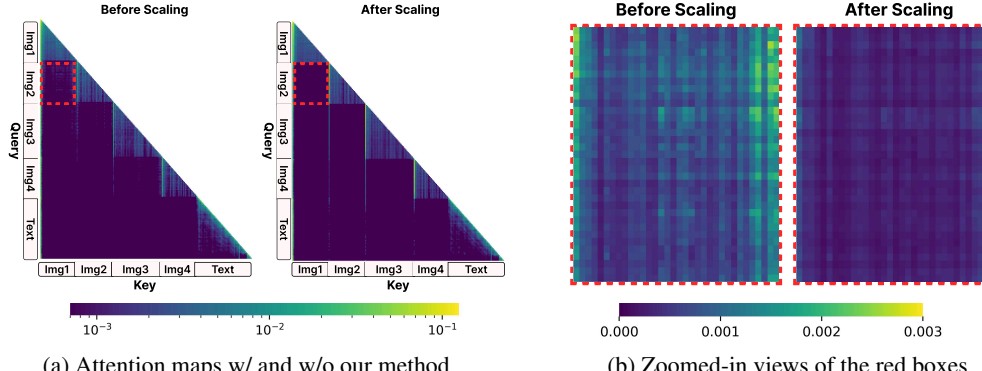

(a) Attention maps w/ and w/o our method   (b) Zoomed-in views of the red boxes

Figure 4: Qualitative comparison of (a) attention maps before and after applying our method, and (b) zoomed-in views of the red boxes. After applying our method, cross-image interaction is reduced.

We effectively reduce cross-image interaction by amplifying this behavior. Due to the normalization effect of the softmax function, emphasizing delimiter tokens reduces the attention allocated to tokens from other images. However, if intra-image interaction were also suppressed, delimiter tokens would no longer fulfill Property 2 (image-tagging effect).

Interestingly, we find that hidden state scaling not only enhances Property 1 but also strengthens image-tagging effect, thereby preserving intra-image interaction. We assume that when scaling is applied, the attention from tokens within an image increases most significantly toward their corresponding delimiter token (see Appendix A.2). According to assumption, this results in tokens from the $i$-th image assigning a higher attention score $p_{d_i}$ to their corresponding delimiter $d_i$ than to other delimiters. In parallel, scaling also increases the magnitude of the value vectors $v_d$ of all delimiter tokens (Figure 3b), which amplifies the contribution of the term $p_{d_i} v_{d_i}$ in the attention output (Figure 3c, red box). As a result, the suppressive effect of the softmax is mitigated, and intra-image attentions are more effectively preserved.

## 5.2 EMPIRICAL EVIDENCE

We empirically validate the effectiveness of our method using Qwen2.5-VL-3B, focusing on both cross-image leakage suppression and intra-image interaction preservation.

**Reduced Cross-Image Information Leakage.** We analyze attention maps to determine whether undesirable cross-image interactions are reduced. To quantify this, we compute the average attention score between all token pairs from different images. For example, the interaction from Image 3 to Image 1 is calculated as the average attention that tokens in Image 3 allocate to tokens in Image 1.

Figures 4a and 4b illustrate a clear reduction in cross-image interaction. In particular, as in Figure 5a, the attention from Image 3 to Image 1 and from Image 3 to Image 2 drops by approximately 50%. These results indicate that our method substantially suppresses undesired interactions across images.

**Preserved Intra-Image Interaction.** To evaluate whether our method preserves intra-image interaction, we compute the attention scores between all token pairs within Image 3. As seen in the rightmost plot of Figure 5a, the interactions within Image 3 remain largely unaffected, indicating that intra-image interaction is preserved.

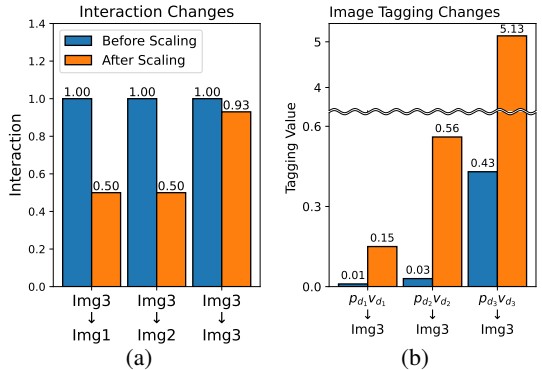

Figure 5: (a) Inter-image interaction changes before and after scaling (normalized to 1.00 before scaling). (b) After scaling, Image 3 tokens receive the strongest attention from the third delimiter token.

Table 1: Performance across four multi-image benchmarks (Mantis, MuirBench, MIRB, QBench2). Applying our method to the Qwen2.5-VL, InternVL3, and LLaVA-OneVision model families generally improves results.

| Dataset | Model | Qwen2.5-VL | | | InternVL3 | | | | LLaVA-OV | |
| | | 3B | 7B | 32B | 1B | 2B | 8B | 14B | 0.5B | 7B |
|---|---|---|---|---|---|---|---|---|---|---|
| Mantis | Baseline | 59.91 | 68.66 | 68.20 | 47.00 | 52.07 | 67.28 | 71.89 | 40.09 | 62.21 |
| | + Ours | **63.13** | **69.12** | **70.05** | **49.77** | **54.38** | **69.12** | **72.81** | **41.01** | **64.06** |
| MuirBench | Baseline | 37.31 | 45.23 | 53.12 | 28.62 | **27.69** | 36.88 | 42.42 | 24.58 | 35.04 |
| | + Ours | **42.42** | **48.15** | **53.82** | **29.38** | 27.65 | **36.92** | **42.58** | **24.85** | **35.35** |
| MIRB | Baseline | 56.45 | **63.57** | 54.90 | 38.49 | 44.38 | 52.32 | 56.45 | 31.79 | 47.88 |
| | + Ours | **57.38** | 63.05 | **55.21** | **40.25** | **46.96** | **52.63** | **57.59** | **32.30** | **48.19** |
| QBench2 | Baseline | 62.70 | 75.80 | 81.40 | **50.80** | 65.20 | 76.50 | 79.60 | 51.70 | 73.90 |
| | + Ours | **63.30** | **76.50** | **81.70** | 50.20 | **65.60** | **76.60** | **80.10** | **51.90** | **74.20** |

Table 2: Results on the WCEP10. R-1, R-2, and R-L denote ROUGE-1, ROUGE-2, and ROUGE-L.

| Model | R-1 | R-2 | R-L |
|---|---|---|---|
| Qwen2.5-3B | 27.30 | 9.75 | 18.42 |
| + Ours | **27.52** | **9.99** | **18.47** |
| Qwen2.5-7B | 29.74 | 11.59 | 20.30 |
| + Ours | **29.77** | **11.70** | **20.35** |
| Phi-1.5 | 9.57 | 1.45 | 7.94 |
| + Ours | **9.80** | **1.49** | **8.09** |

Table 3: Results on the MultiNews. R-1, R-2, and R-L denote ROUGE-1, ROUGE-2, and ROUGE-L.

| Model | R-1 | R-2 | R-L |
|---|---|---|---|
| Qwen2.5-3B | 37.16 | 10.85 | 18.81 |
| + Ours | **37.24** | **10.90** | **18.84** |
| Qwen2.5-7B | 37.18 | 11.26 | 19.15 |
| + Ours | **37.19** | **11.29** | **19.17** |
| Phi-1.5 | 26.30 | 5.73 | 14.55 |
| + Ours | **26.36** | **5.76** | **14.61** |

This preservation can be attributed to the reinforced image-tagging behavior of delimiter tokens. To support this claim, we compute the average attention output received by all tokens in Image 3 from each delimiter token. Figure 5b shows that Image 3 tokens receive the strongest attention output from the third delimiter token, confirming that image tagging is enhanced and helps maintain intra-image interaction.

## 5.3 DISCUSSIONS

**Computational Benefits.** Our method modifies hidden states without altering the attention mechanism, allowing compatibility with optimized attention kernels such as FlashAttention (Dao, 2023). In contrast, modifying attention weights directly would disrupt these optimizations and significantly increase memory usage, especially in multi-image inputs. For example on the MIRB dataset, inference with Qwen2.5-VL 3B model fails due to memory constraints even with 140GB VRAM when attention is modified. In contrast, our method using FlashAttention runs successfully with the 32B model and highlights its efficiency.

**Preservation of Text-Image Interaction.** Enhancing image tagging behavior may raise concerns about interfering with text-vision interactions. However, our experiments show that such effects are minimal. Text tokens are already known to receive strong mutual attention (Chen et al., 2024) and are largely unaffected by the strengthened delimiter tokens. In practice, text-to-image interaction scores drop by only 10%, and the overall interaction between modalities remains robust, indicating that text-vision alignment is well preserved.

## 6 EXPERIMENTS

### 6.1 BENCHMARKS AND SETTINGS

**Multi-Image Benchmarks.** We applied our method to four multi-image benchmarks to evaluate its effectiveness. Mantis-Eval (Jiang et al., 2024) is a benchmark suite designed to evaluate multi-image capabilities, consisting of 8 multi-image benchmarks and 6 single-image benchmarks that test

Table 4: Accuracy on the TQABench dataset for Qwen2.5 models.

| Model | Accuracy |
|---|---|
| Qwen2.5-3B | 37.38 |
| + Ours | **37.84** |
| Qwen2.5-7B | 37.50 |
| + Ours | **38.14** |

Table 5: Ablation results on delimiter tokens, M-RoPE, and our method.

| Delim | M-RoPE | Ours | Accuracy |
|---|---|---|---|
| ✔ | ✘ | ✘ | 59.91 |
| ✘ | ✔ | ✘ | 53.92 |
| ✔ | ✔ | ✘ | 62.21 |
| ✔ | ✘ | ✔ | **63.13** |

various skills such as co-reference, comparison, and temporal reasoning. MuirBench (Wang et al., 2024a) evaluates 12 types of multi-image understanding with 2,600 questions over 11,264 images, covering spatial, diagram, and retrieval tasks. MIRB (Zhao et al., 2024) is a benchmark designed to evaluate LVLMs' ability to compare, analyze, and reason across multiple images, covering perception, world knowledge, reasoning, and multi-hop reasoning. Q-Bench2 (Zhang et al., 2024) is a benchmark designed to evaluate the low-level visual perception abilities of MLLMs, extending beyond a single image to include image pairs. It specifically assesses cross-image reasoning and human-like comparative judgment by testing models on pairwise visual inputs.

**Multi-Document and Multi-Table Benchmarks.** Motivated by the idea that our approach could generalize to other multi-instance settings, we further evaluate it on multi-document and multi-table benchmarks. Specifically, we applied our method to MultiNews and WCEP-10 for multi-document benchmarks, and to TQABench for the multi-table benchmark. MultiNews (Fabbri et al., 2019) is a large-scale dataset for multi-document summarization, consisting of clusters of news articles and corresponding human-written summaries. WCEP10 (Ghalandari et al., 2020) is a multi-document summarization dataset pairing news article clusters with short human-written summaries from the Wikipedia Current Events Portal. TQABench (Qiu et al., 2024) is a multi-table QA benchmark designed to assess LLMs' ability to handle complex question answering over relational data. We applied our method to the 8k split of TQABench.

**Implementation Details.** For the multi-image tasks, we used Qwen2.5-VL (3B, 7B, 32B) (Bai et al., 2025), InternVL3 (1B, 2B, 8B, 14B) (Zhu et al., 2025), and LLaVA-OneVision (0.5B, 7B) (Li et al., 2024) as the vision-language models. For the multi-table task, we employed Qwen2.5 (3B, 7B) (Team, 2024), and for the multi-document task, we used Qwen2.5 (3B, 7B) and Phi-1.5 (Li et al., 2023b). All other details are in Appendix A.3.

## 6.2 RESULTS

**Results on Multi-Image Understanding.** As shown in Table 1, our method consistently improves performance across all model families, including Qwen2.5-VL, InternVL3, and LLaVA-OneVision. The improvements are observed across a wide range of benchmarks such as Mantis, Muirbench, MIRB, and Qbench2, demonstrating the robustness of our approach. For example, on the Muirbench benchmark, the Qwen2.5-VL-3B model improves from 37.31 to 42.42, and the InternVL3-2B model improves from 52.07 to 54.38 on Mantis. Notably, performance gains appear across models of various sizes, from small-scale (e.g., 0.5B) to large-scale (e.g., 32B), indicating that the proposed delimiter token scaling method is effective regardless of model capacity. These consistent improvements across diverse models and benchmarks for multi-image understanding highlight the generality and practicality of our method.

**Results on Multi-Document and Multi-Table Understanding.** Tables 2 and 3 present the ROUGE scores on multi-document summarization tasks. On both the WCEP10 and MultiNews datasets, the proposed delimiter token scaling method consistently improves ROUGE-1, ROUGE-2, and ROUGE-L scores across all models. Similar improvements are observed in both the Qwen2.5-7B and Phi-1.5 models. Table 4 further shows consistent gains on the multi-table reasoning benchmark, TQABench. Notably, the Qwen2.5-3B model with our method even outperforms the 7B baseline, which is a compelling result. This indicates that our delimiter token scaling method can yield meaningful performance gains beyond what can be achieved by increasing model size. These

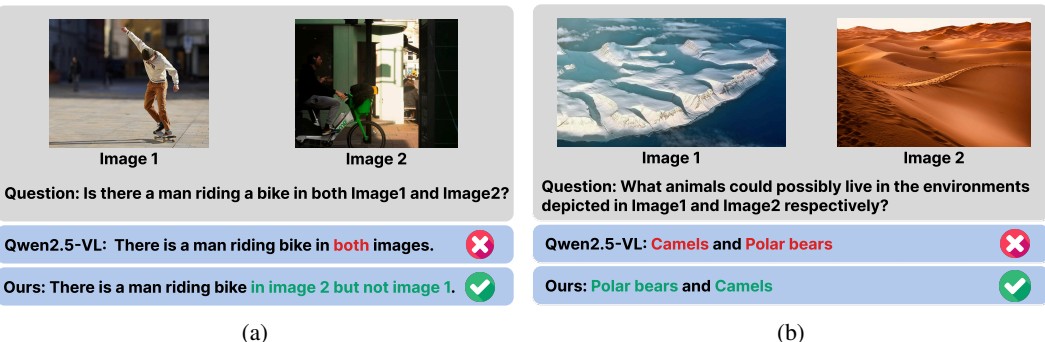

Figure 6: Qualitative results on the Mantis benchmark. Although the tasks are multi-choice, answers are shown in sentence form to show that our method reduces cross-image leakage while the baseline Qwen2.5-VL fails.

results demonstrate that our method is broadly applicable across different input modalities, not just limited to multi-image settings.

**Qualitative Results.** We qualitatively analyze the model outputs in Figure 6. In Figure 6a, the baseline model incorrectly states that both images contain a man riding a bicycle, although in reality, only the second image contains this. This shows a case of cross-image information leakage, where the information from the second image contaminates the understanding of the first. In contrast, our method enables the model to correctly identify that only the second image contains the man on a bicycle. In Figure 6b, the correct answer is "polar bears and camels", with each animal appearing in a different image. However, the baseline model returns "camels and polar bears", reversing the correspondence. With our method, the model preserves the distinction between the two images and produces the correct answer. These examples show that our method effectively reduces cross-image information leakage, leading to more accurate and disentangled reasoning across multiple images.

**Comparison with M-RoPE.** In Qwen2-VL (Wang et al., 2024b), temporal positional embeddings are applied to video frames to distinguish them along the temporal axis. This is conceptually similar to our image-specific tagging method. Motivated by this, we conducted a comparative experiment with the M-RoPE-based temporal embedding method, where temporal positional embeddings were injected into each image.

As shown in Table 5, applying M-RoPE alone results in lower performance than the baseline. When combined with image delimiter tokens, M-RoPE improves performance beyond the baseline but still lags behind our method. These findings suggest that introducing mechanisms to help the model better distinguish between images—such as M-RoPE or delimiter tokens—can mitigate performance degradation caused by cross-image information leakage. Notably, although M-RoPE was originally designed for temporal distinction in video tasks, it also improves performance in multi-image settings. This further supports our hypothesis that insufficient image distinction is a key cause of performance drops. Overall, these results indicate our simple hidden state scaling approach can be more effective at resolving such confusion than more complex temporal embedding strategies.

**Few-Shot Evaluation with Interleaved Examples.** We additionally conducted few-shot evaluations. We reorganized the single-image dataset into a few-shot setting by constructing 4-shot interleaved inputs, where each image is followed in order by its corresponding question and answer. Evaluating this setup on the validation-lite (lmms-lab/LMMs Eval-Lite) splits of TextVQA (Singh et al., 2019) and OKVQA (Marino et al., 2019), we observed consistent performance improvements across Qwen2.5-VL-3B, Qwen2.5-VL-7B, and InternVL3-8B, as shown in the Table 6. Since this task requires understanding both the example images and the accompanying text, image–text interaction is crucial. The improved performance, demonstrating

Table 6: Few-shot Performance on OKVQA and VizWiz.

| Dataset | Model | Qwen2.5-VL | | Intern |
| --- | --- | --- | --- | --- |
| | | 3B | 7B | 8B |
| OKVQA | Baseline | 18.04 | 27.56 | 46.84 |
| | + Ours | **20.00** | **28.24** | **48.68** |
| VizWiz | Baseline | 42.38 | 53.70 | 47.04 |
| | + Ours | **42.88** | **54.36** | **50.92** |

Table 7: Performance on the Mantis Benchmark for Large Models.

| Method | Qwen2.5-VL 72B | InternVL3 78B |
|---|---|---|
| Baseline | 74.19 | 74.65 |
| + Ours | **75.58** | **76.50** |

Table 8: Our method introduces no additional cost in memory (GB) or inference time (s).

| | Avg Memory | Peak Memory | Inference Time |
|---|---|---|---|
| Baseline | $8.3 \pm 0$ | $10.2 \pm 0$ | $100 \pm 1$ |
| Ours | $8.3 \pm 0$ | $10.2 \pm 0$ | $99.7 \pm 0.6$ |

that our method can also be effectively applied to downstream tasks where the relationship between images and text plays an important role. These findings further indicate that our approach is applicable to interleaved data and remains effective in few-shot settings, showing that it can generalize beyond the originally tested specialized input format and extend to a broader range of scenarios.

**Performance on Larger-Scale Models.** We conducted additional experiments on even larger models. We evaluated our method on the Mantis benchmark using Qwen2.5-VL-72B and InternVL3-78B, which represent the largest models in the Qwen2.5-VL and InternVL3 families, respectively. As shown in the Table 7, the results show that both models exhibit performance improvements when applying our method. This indicates that our approach remains effective as model scale increases and can be reliably applied to extremely large-scale models.

**Inference Cost.** We empirically verified that our method introduces no additional inference cost. Using four GPUs, we repeated the experiment three times and averaged the results. As shown in the Table 8, both the average and peak VRAM usage were identical to the Baseline, and the inference time also remained unchanged. These results demonstrate that, in practice, our method does not incur any additional inference overhead. Our approach is fully compatible with FlashAttention, as it operates at the hidden-state level without requiring any direct modification to the attention computation.

**Hyperparameter Sensitivity.** As shown in Figure 7, we experimented with a range of scaling values for the $\lambda$ and analyzed their impact. The results demonstrate consistent performance improvements across most settings compared to the baseline, indicated by the red dashed line, showing that our method is robust to variations in this hyperparameter. These findings support the idea that appropriately amplifying the hidden states of image delimiter tokens effectively mitigates cross-image information leakage.

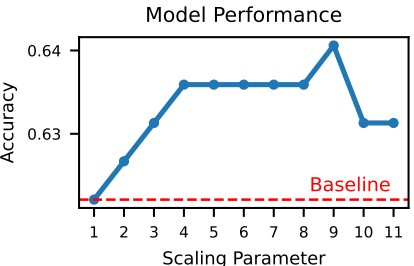

Figure 7: Sensitivity on hyperparameter $\lambda$.

## 7 LIMITATION

Our method is currently not applicable to videos that lack explicit frame-separating tokens. In future work, we plan to extend our approach to the video domain by incorporating mechanisms to model temporal transitions between frames, which are crucial for understanding dynamic visual content.

## 8 CONCLUSION

In this work, we address the issue of cross-image information leakage in multi-image input settings by analyzing the role and limitations of image delimiter tokens, which are responsible for separating visual inputs. Based on this analysis, we propose a simple method that enhances the functionality of these tokens, effectively suppressing interactions across different images while preserving interactions within the same image. Our method consistently improves performance across various multi-image benchmarks and also demonstrates its generalizability to text-only settings, such as multi-document and multi-table understanding. It is easy to integrate and introduces no additional training or inference cost.

**Ethics Statement.** We use only publicly available datasets and do not involve human subjects or sensitive data. The method is training-free and efficient, with no foreseeable ethical concerns beyond responsible use.

**Reproducibility Statement.** We will release full code and scripts. All experiments were run with fixed seeds, and datasets and hyperparameters are documented to ensure reproducibility.

## ACKNOWLEDGEMENTS

We are thankful to Beomyun Kwon, Jimin Hong, Elena Kuular and Arnas Uselis for their thoughtful discussions and constructive comments. This work was supported by the Ministry of Education of the Republic of Korea and the National Research Foundation of Korea (NRF-2024S1A5C3A03046168, Contribution: 50%) and by the Institute of Information & Communications Technology Planning & Evaluation (IITP) grant funded by the Korea government (MSIT) (RS-2025-25461932, Elite Research-driven Technology Development for Advanced Large-Scale LLM/VLMs and ASEAN Language Expansion, Contribution: 50%).

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

# A    APPENDIX

## A.1    ROLE OF IMAGE-DELIMITER TOKENS

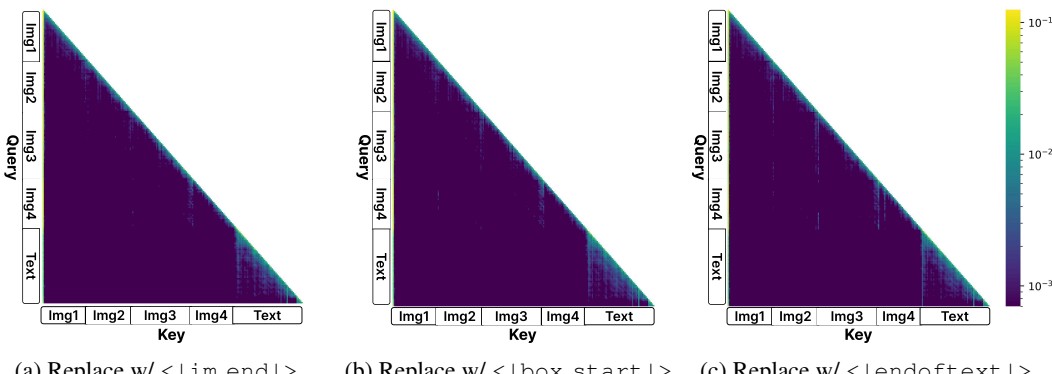

(a) Replace w/ `<|im_end|>`    (b) Replace w/ `<|box_start|>`    (c) Replace w/ `<|endoftext|>`

Figure A1: Replacing `<|vision_start|>` with other tokens (e.g., `<|im_end|>`, `<|box_start|>`, `<|endoftext|>`) fails to separate images effectively.

We additionally replaced the image delimiter token with various other special tokens beyond `<|im_start|>`, including `<|im_end|>`, `<|box_start|>`, and `<|endoftext|>`, and analyzed the resulting attention maps. As shown in the Figure A1, these tokens exhibited trends similar to `<|im_start|>`, suggesting that special tokens like `<|vision_start|>` effectively serve as image delimiters that separate visual content. This observation was also reflected in the performance results. Removing the image delimiter tokens or replacing them

Table A1: Effect of removing or replacing image delimiter tokens with other special tokens.

| Delim | Special Token | Accuracy |
|:---:|:---:|:---:|
| ✔ | ✘ | 59.91 |
| ✘ | ✔ | 53.92 |
| ✘ | ✘ | 53.46 |

with other special tokens led to a performance drop. As shown in Table A1, we can clearly observe this degradation in performance. The sample used in this analysis (i.e., the four images and the corresponding text) is provided in A6, while additional example attention maps are included in Section A.8.

## A.2    EMPIRICAL EVIDENCE OF ASSUMPTION

We previously assumed that tokens within each image would show the greatest increase in attention toward their corresponding image delimiter token. In this section, we empirically validate this assumption by analyzing the attention score increases for each delimiter token when the query token belongs to the third image. The third delimiter token, which corresponds to the third image, exhibits the most significant increase, approximately 52 times greater than the first delimiter token and 9 times greater than the second. These findings confirm that attention concentrates most strongly on the aligned delimiter token.

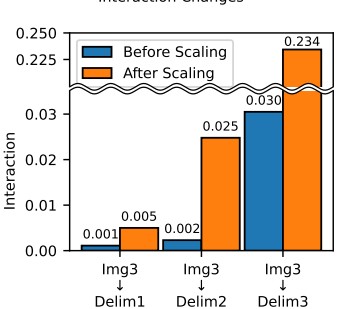

Figure A2: Tokens in Image 3 show the largest attention increase to its delimiter.

## A.3    EXPERIMENTAL SETTINGS

For the multi-image tasks, we use 10% of the test set as a validation set to determine hyperparameters such as the scaling layer and scaling factor. In contrast, for multi-document and multi-table tasks, we search hyperparameters directly on the test set without a separate validation split. For all reported results, we tuned scaling hyperparameter for each model to identify and apply the optimal value. To avoid excessive tuning, we fix the selected scaling layer for each model and use it consistently across all benchmarks. Details are provided in the code.

Table A3: First Token Scaling Ablation

| Method | Accuracy |
|---|---|
| Baseline | 59.91 |
| First token scaling | 60.37 |
| Ours | **63.13** |

Table A4: Delimiter Replacement Ablation

| Method | Accuracy |
|---|---|
| Baseline | 59.91 |
| `<|im_start|>` Scaling | 53.00 |
| Ours | **63.13** |

When applying delimiter token scaling, we use task-specific special tokens to distinguish input units such as images, documents, and tables.

In multi-image benchmarks, we apply delimiter token scaling to the model-specific special tokens that separate each image. For Qwen2.5-VL, we use `<|vision_start|>` and `<|vision_end|>`, for InternVL3, `` and `<\img>`, and for LLaVA-OneVision, line breaks (`\n`).

In multi-document and multi-table settings, we designate appropriate separator tokens as special delimiters to split individual documents or tables within the input sequence. For example, in the MultiNews dataset, documents are separated using the token `||||||`, and we designate this token as a special delimiter to apply our method.

All experiments are conducted on NVIDIA GPUs, including A5000, A6000, and H200, depending on resource requirements.

## A.4 ABLATION STUDY

**Comparing scaling applied individually to Q, K, and V.** When the hidden state is scaled, it is subsequently transformed through the projection layers into Q, K, and V, meaning that the scaling effect directly influences all three components. Based on this observation, we conducted an ablation study to analyze how the scaled hidden state affects each component when applied independently. For each of Q, K, and V, we applied scaling after the corresponding projection. As shown in the Table A2 results show that applying scaling to any single component—Query, Key, or Value—leads to performance improvements over the Baseline. Notably, scaling K yields a larger performance gain than the Q-only or V-only settings. This is because increasing the Key of the delimiter token makes the Queries match more strongly with it, which then produces a clearer block-wise attention pattern across the image chunks. However, this effect alone is insufficient to induce meaningful image tagging. Our method, which applies scaling to Q, K, and V simultaneously, achieves the highest performance. This demonstrates that Q, K, and V each contribute to strengthening attention and establishing the image-tagging structure, and that jointly scaling all three components is most effective for multi-image understanding. These results are based on experiments conducted on the Qwen2.5-VL-3B model using the Mantis Benchmark.

Table A2: Ablation on Query, Key, Value Scaling.

| Method | Accuracy |
|---|---|
| Baseline | 59.91 |
| Q scaling | 61.75 |
| K scaling | 62.67 |
| V scaling | 61.29 |
| Ours | **63.13** |

**Scaling an Alternative Special Token.** We observed that delimiter tokens inherently distinguish images to some extent, and we proposed delimiter token scaling to further enhance this effect. To verify this, we conducted two ablation studies: 1) keeping the delimiter tokens unchanged while scaling the first token, which is known to receive strong attention, and 2) replacing the delimiter tokens with other special tokens and applying the scaling method to those alternatives. All experiments were conducted on the Mantis Benchmark using the Qwen2.5-VL-3B model.

When we scaled the first token, `<|im_start|>`, as shown in the Table A3, the performance improved slightly compared to the Baseline but remained substantially lower than Ours. We consider that this small improvement arises because the first token is a sink token; scaling it further amplifies its already dominant attention, allowing it to absorb some cross-image interactions However, because this approach cannot provide any image-level tagging effect, its performance is significantly inferior to our method.

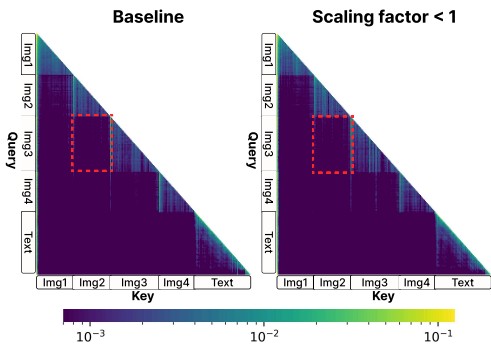 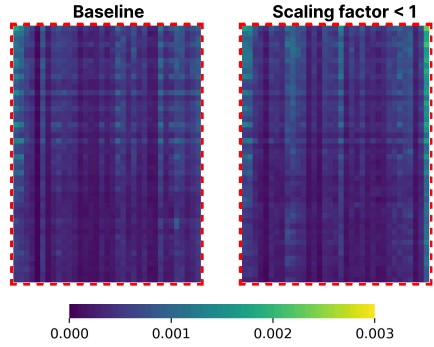

(a) Atten maps of baseline and scaling factor<1      (b) Zoomed-in views of the red boxes

Figure A4: Qualitative comparison of (a) attention maps before and after applying a scaling factor smaller than 1, and (b) zoomed-in views of the red boxes. With a scaling factor below 1, cross-image interaction is still present.

Next, when we replaced the delimiter token with the `<|im_start|>` token and applied scaling in the same manner, as shown in the Table A4, the performance dropped below the Baseline. This indicates that scaling a token that merely appears between images does not yield any performance gain; only scaling the actual delimiter token leads to improvements.

Together, these experiments demonstrate that performance does not improve by simply scaling an arbitrary token. Instead, the effectiveness of our method arises specifically from the structural and functional role that the delimiter token plays in multimodal models.

**Scaling with a value smaller than 1.** Our method applies a scaling factor greater than 1 to the delimiter token. To perform an ablation study on the scaling behavior, we additionally conducted experiments in which the scaling factor was set to a value smaller than 1. When the scaling factor is less than 1, the delimiter token is expected to receive insufficient attention, making it difficult to effectively suppress cross-image information leakage. This experiment was conducted on the Mantis Benchmark using the Qwen2.5-VL-3B model. As shown in the Figure A3, the performance drops below

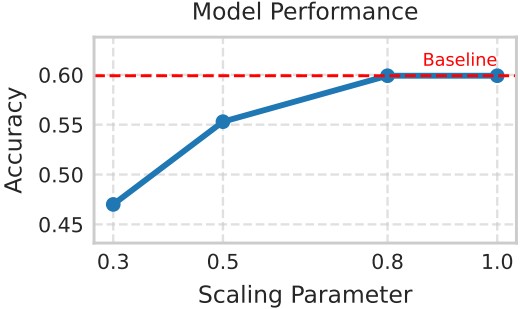

Figure A3: Scaling smaller value

the Baseline. The attention map analysis in the Figure A4 also reveals that cross-image information leakage persists and, in some regions, becomes even more pronounced. We attribute this behavior to the reduced scaling factor, which prevents the delimiter token from receiving sufficiently strong attention.

**Layer Selection.** Layer selection was determined through hyperparameter tuning, and for the same model, we kept the selected layers consistent even when the benchmark changed. In some models, only a single layer was chosen, while in others, multiple layers were selected. Guided by prior study (Yu & Lee, 2025) reporting that early layers play an important role in grounding and other vision-related processing, we primarily selected consecutive early layers. Moreover,

Table A5: Layer Selection Ablation

| Select Layer | Accuracy |
|---|---|
| Baseline | 0.5991 |
| **0,1,2,3** | **0.6313** |
| 10,11,12,13 | 0.5991 |
| 22,23,24,25 | 0.5484 |
| 32,33,34,35 | 0.5945 |

since the scaling applied in early layers propagates through subsequent layers, we considered early-layer selection to be more effective. We validated the performance differences arising from layer selection through experiments. Using the Mantis benchmark and the Qwen2.5-VL-3B model, which consists of 36 layers (0–35), we selected layers 0, 1, 2, and 3. As shown in the Table A5, selecting early layers yields the best performance.

Table A6: Comparison with FOCUS.

| Method | Qwen2.5-VL | | InternVL3 | |
|---|---|---|---|---|
| | 3B | 7B | 1B | 8B |
| Baseline | 59.91 | 68.66 | 47.00 | 67.28 |
| FOCUS | 58.53 | – | 47.93 | – |
| Ours | **63.13** | **69.12** | **49.77** | **69.12** |

Table A7: Memory and runtime comparison.

| Method | Avg VRAM | Peak VRAM | Inference Time |
|---|---|---|---|
| Baseline | 8.27 GB | 10.18 GB | 1m 40s |
| FOCUS | 10.76 GB | 21.86 GB | 5m 21s |
| Ours | 8.27 GB | 10.18 GB | 1m 41s |

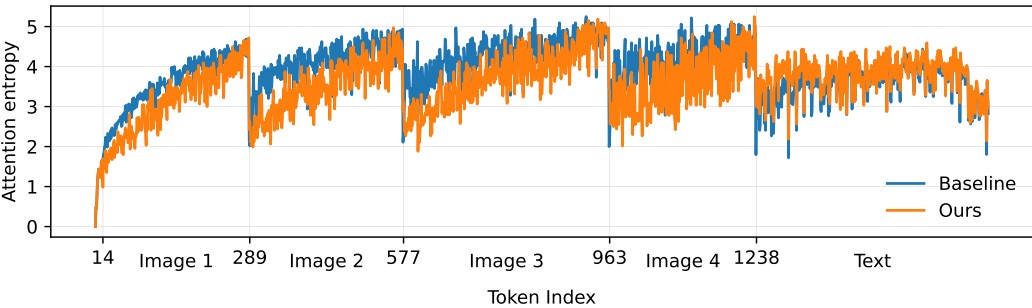

Figure A5: Delimiter tokens exhibit lower entropy and our method further reduces entropy in the image region by suppressing cross-image attention.

## A.5 COMPARISON WITH FOCUS.

We compared our method with FOCUS (Park et al., 2025), a previous method designed to mitigate cross-image information leakage. For fairness, we performed a grid search over the hyperparameters, which resulted in a total of 81 configurations, and we used the best-performing one for comparison. As shown in the Table A6, our method consistently outperforms FOCUS on the Mantis Benchmark. Table A7 further reveals that FOCUS incurs substantially higher memory consumption, to the extent that certain configurations could not be executed on Qwen2.5-VL-7B and InternVL3-8B due to memory limitations. In contrast, our approach is markedly more memory-efficient, requiring roughly half the peak VRAM usage of FOCUS, while also achieving better runtime efficiency.

## A.6 ANALYSIS OF ATTENTION ENTROPY CHANGES

We also conducted an attention entropy analysis to further examine how our method affects cross-image attention. We first measured token-level attention entropy in the baseline setting with multi-image inputs and then compared how the entropy changes when our method is applied. As shown in Figure A5, the baseline exhibits a sharp drop in entropy at the delimiter position between images. This behavior matches the attention patterns in Figure 1a. For example, immediately before the delimiter of Image 3 (i.e., at the final Query position of Image 2), strong intra-image interaction leads to higher entropy. Once the delimiter token of Image 3 becomes the Query token, however, attention to Images 1 and 2 largely disappears and concentrates on the sink token and the delimiter itself, resulting in lower entropy. In short, the delimiter token absorbs much of the attention and suppresses backward cross-image attention, reducing entropy. At the same time, as highlighted by the red box in Figure 1a, delimiter tokens in the baseline still allow a non-negligible amount of cross-image attention. In contrast, Figure 4 shows that our method substantially reduces these cross-image interactions. Accordingly, we expected the entropy in the image region to decrease when applying our method. This is confirmed by the orange curve in Figure A5: entropy in the image region is consistently lower than in the baseline, whereas entropy in the text region remains nearly unchanged, indicating that text–image interactions are preserved. These findings are consistent with our earlier analysis: our method suppresses cross-image information leakage while maintaining text–image interaction, and this effect is clearly reflected in the entropy patterns.

Table A8: Performance with the entropy-based extension.

| Method | InternVL3 14B | Llava-OV 0.5B |
|---|---|---|
| Baseline | 71.89 | 40.09 |
| Ours | 72.81 | 41.01 |
| + Entropy | **73.27** | **41.47** |

Table A9: The entropy-based extension adds extra cost in both memory usage (GB) and inference time (s/input).

| Method | Inference Time | Peak Memory | Avg Memory |
|---|---|---|---|
| Baseline | 0.77 | 60.7 | 60.0 |
| Ours | 0.77 | 60.7 | 60.0 |
| + Entropy | **1.78** | **114.8** | **95.7** |

## A.7 FURTHER EXTEND OUR METHOD THROUGH ENTROPY

We extended our original method by incorporating an additional mechanism that leverages attention entropy. Specifically, we dynamically determined the scaling parameter by computing the entropy of each delimiter token's attention-weight distribution. When the entropy was higher—i.e., when the attention was more dispersed—we applied stronger scaling to the corresponding delimiter's hidden state. As shown in the Table A8, this extended mechanism provides additional performance gains on the Mantis Benchmark when using the InternVL3-14B model. Following the same evaluation protocol as the Baseline and Ours settings, all experiments were conducted on the test set, with 10% of the test set used as a validation split. However, a critical drawback of this method is that it requires direct access to the full attention map for entropy computation, which prevents the use of FlashAttention. As shown in the Table A9, the inference speed becomes more than twice as slow compared to our original method, and the memory overhead is also substantial. The average GPU memory consumption increases by approximately $1.6\times$, requiring an additional 36GB—already exceeding the capacity of a single 24GB GPU. Moreover, the peak memory usage increases by about 54GB, effectively requiring more than two additional 24GB GPUs, making the method highly impractical in real-world scenarios.

## A.8 ADDITIONAL QUALITATIVE RESULTS

In addition to the original sample, we analyzed several additional cases, and the results are presented in Figures A7, A8, A9, A10 and A11 In particular, these two samples illustrate situations where the input images are visually very similar to each other. Even in such challenging settings, the results remain consistent. When the delimiter token is present, the images are clearly separated, forming a triangular boundary pattern. In contrast, when the delimiter token is removed or replaced with another special token such as <|im_start|>, the attention maps no longer show clear separation between images. These findings indicate that the delimiter token plays a crucial role in distinguishing images, even when they are visually similar.

## A.9 ADDITIONAL LIMITATION

Our method requires modifying hidden states, and in its current form, this imposes a limitation in that it is applicable only to open-source models. Although not directly applicable to proprietary models, our simple and lightweight reweighting mechanism can be easily integrated into such systems. Although external users of commercial models cannot access hidden states, for model developers, incorporating our method internally poses little technical difficulty. Thus, rather than being a feature limited to open-source models, our approach can be seen as an intuitive improvement that is reasonably applicable to commercial systems as well.

## A.10 ADDITIONAL DISCUSSION

Recent studies in the LLM literature have examined the behavior of models under parallel context encoding, where multiple interleaved contexts are presented within a single input sequence. These works highlight that sink tokens often exhibit abnormal hidden-state activations under such settings, which in turn lead to increased and irregular attention entropy. Prior analyses further argue that the model encounters a multi-sink pattern at inference that it has never observed during training, and that this distributional shift directly contributes to performance degradation.

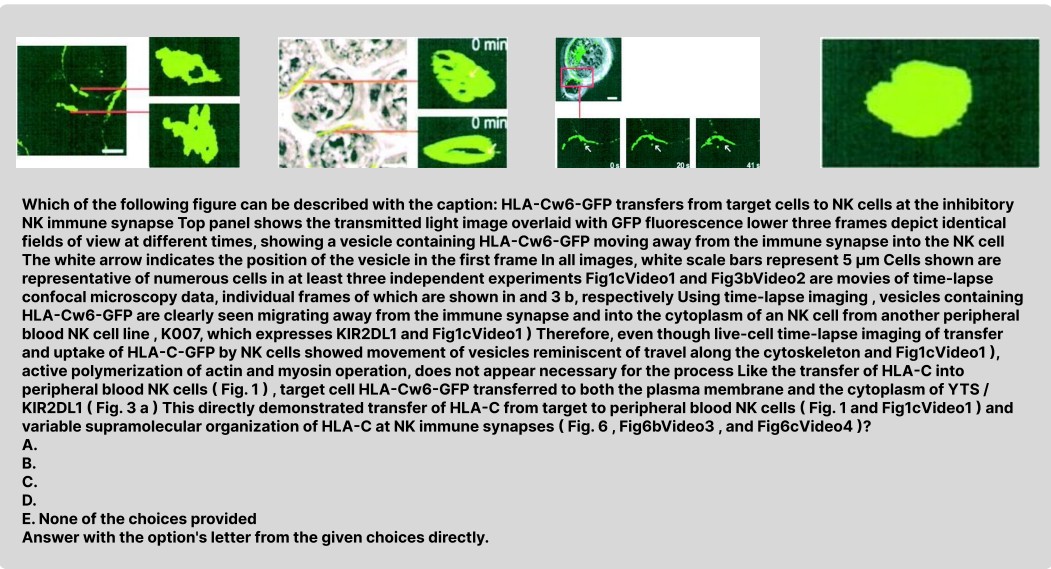

Figure A6: Input sample consisting of the images and the query used for the attention map in Section A.1.

Our work shares part of this motivation, as we also investigate how a model processes and regulates multiple interleaved segments—such as multi-image inputs—within one sequence. However, the setting we study differs from the parallel context encoding scenario in several fundamental ways.

First, prior work focuses on LLMs, whereas our study centers on LVLMs, whose input structure and training signals differ substantially. Second, the types of patterns the models observe during training, as well as the phenomena that arise at inference, diverge meaningfully between the two settings. In the parallel context encoding literature, the abnormal hidden states of sink tokens are considered a root cause of unstable attention behavior, and performance deterioration is attributed to the model's first exposure to a previously unseen multi-sink pattern during inference.

In contrast, our analysis (see Figure 1a) shows that LVLMs naturally experience multi-sink–like patterns during training due to their reliance on delimiter tokens. Because these delimiter tokens introduce repeated structural cues throughout the training corpus, LVLMs are routinely exposed to patterns that closely resemble the multi-sink configuration. As a result, LVLMs do not face a novel or unexpected pattern at inference time; rather, they encounter a familiar structural arrangement that they have already internalized during training.

For this reason, even if attention entropy increases to some extent in LVLMs, such changes are unlikely to directly cause performance degradation. This distinction highlights that the mechanisms underlying multi-context processing in LVLMs differ substantially from those observed in LLM-based parallel context encoding, owing largely to LVLMs' structural use of delimiter tokens and the training distributions they induce.

### A.11 BROADER IMPACT

Our method improves efficiency and sustainability by enhancing performance without additional training or inference costs, reducing reliance on large datasets and retraining. This lowers energy use and research expenses, contributing to a smaller carbon footprint.

### A.12 THE USE OF LARGE LANGUAGE MODELS (LLMS)

We used a large language model (ChatGPT) to improve the clarity and fluency of the manuscript. All technical ideas, analyses, and results were solely developed by the authors.

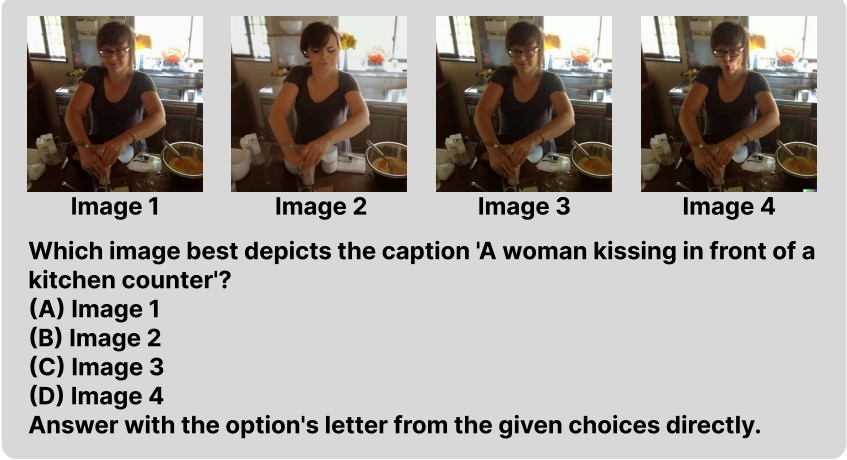

(a) Input sample consisting of the images and the query used for the attention map.

(b) Baseline

(c) Ours

(d) w/ delimiter tokens

(e) w/o delimiter tokens

(f) Replace w/ `<|im_start|>`.

(g) Replace w/ `<|im_end|>`.

(h) Replace w/ `<|box_start|>`.

(i) Replace w/ `<|endoftext|>`.

Figure A7: Visualization of attention patterns for the bottled wine example. We compare the baseline model, our delimiter-token scaling method, and ablations where the image delimiter token is removed or replaced with other special tokens.

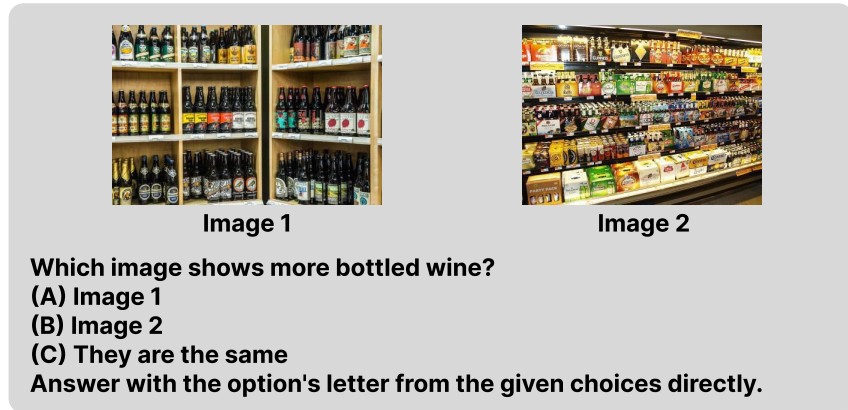

(a) Input sample consisting of the images and the query used for the attention map.

(b) Baseline

(c) Ours

(d) w/ delimiter tokens

(e) w/o delimiter tokens

(f) Replace w/ `<|im_start|>`.

(g) Replace w/ `<|im_end|>`.

(h) Replace w/ `<|box_start|>`.

(i) Replace w/ `<|endoftext|>`.

Figure A8: Visualization of attention patterns for the bottled wine example. We compare the baseline model, our delimiter-token scaling method, and ablations where the image delimiter token is removed or replaced with other special tokens.

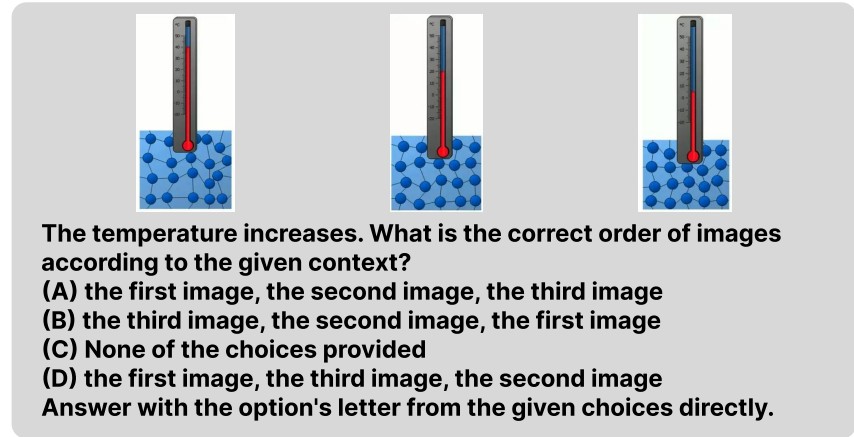

(a) Input sample consisting of the images and the query used for the attention map.

(b) Baseline

(c) Ours

(d) w/ delimiter tokens

(e) w/o delimiter tokens

(f) Replace w/ `<|im_start|>`.

(g) Replace w/ `<|im_end|>`.

(h) Replace w/ `<|box_start|>`.

(i) Replace w/ `<|endoftext|>`.

Figure A9: Visualization of attention patterns for the bottled wine example. We compare the baseline model, our delimiter-token scaling method, and ablations where the image delimiter token is removed or replaced with other special tokens.

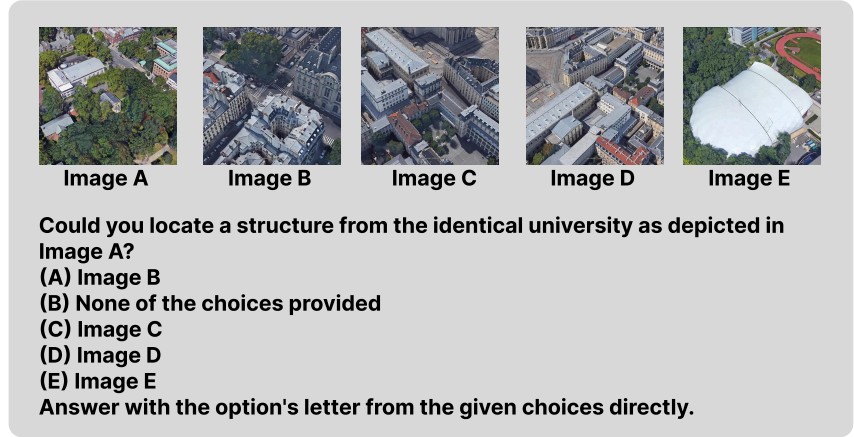

(a) Input sample consisting of the images and the query used for the attention map.

(b) Baseline

(c) Ours

(d) w/ delimiter tokens

(e) w/o delimiter tokens

(f) Replace w/ `<|im_start|>`.

(g) Replace w/ `<|im_end|>`.

(h) Replace w/ `<|box_start|>`.

(i) Replace w/ `<|endoftext|>`.

Figure A10: Visualization of attention patterns for the bottled wine example. We compare the baseline model, our delimiter-token scaling method, and ablations where the image delimiter token is removed or replaced with other special tokens.

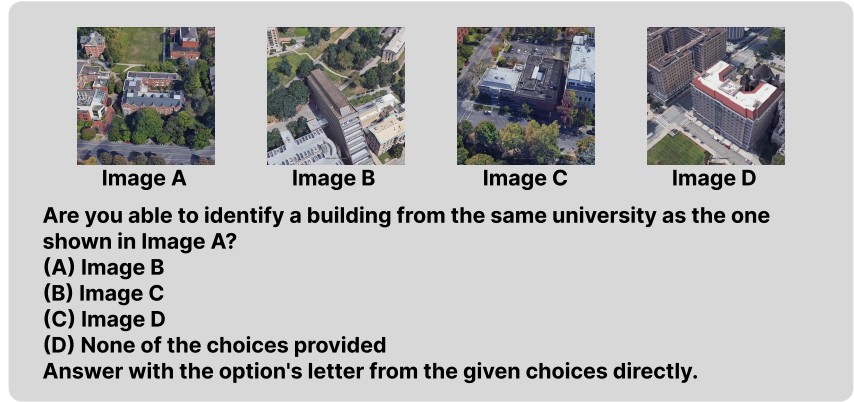

(a) Input sample consisting of the images and the query used for the attention map.

(b) Baseline

(c) Ours

(d) w/ delimiter tokens

(e) w/o delimiter tokens

(f) Replace w/ `<|im_start|>`.

(g) Replace w/ `<|im_end|>`.

(h) Replace w/ `<|box_start|>`.

(i) Replace w/ `<|endoftext|>`.

Figure A11: Visualization of attention patterns for the bottled wine example. We compare the baseline model, our delimiter-token scaling method, and ablations where the image delimiter token is removed or replaced with other special tokens.

