# OpenReview forum: "Enhancing Multi-Image Understanding through Delimiter Token Scaling"
_ICLR.cc/2026/Conference — ICLR 2026 Poster_

### Official Review · Reviewer_Fgst · 2025-10-27

**Soundness:** 3
**Presentation:** 3
**Contribution:** 3
**Rating:** 6
**Confidence:** 4

**Summary:**

The method is evaluated on a wide range of models and benchmarks, demonstrating consistent performance improvements on multi-image understanding tasks (Mantis, MuirBench, etc.). Crucially, the authors show the method's generality by applying it to text-only multi-document and multi-table tasks, again achieving performance gains. The proposed approach requires no additional training or inference costs.

**Strengths:**

The paper's primary strength is its clear and insightful analysis of how delimiter tokens function and where they fall short. The "image-wise tagging" concept provides a strong theoretical motivation for the proposed solution.

The method is remarkably simple (a single scaling operation) and highly efficient (no training, no inference overhead, compatible with optimizations like FlashAttention). This makes it a very appealing and practical technique.

The effectiveness of the method is demonstrated across a wide variety of models (Qwen2.5-VL, InternVL3, LLaVA-OV), model sizes (0.5B to 32B), and task domains (multi-image, multi-document, multi-table). This shows it is a general principle for improving multi-instance understanding, not a model-specific trick.

**Weaknesses:**

The paper states that the scaling layer and factor λ are tuned for each model. The appendix mentions the scaling layer is fixed for each model, but the process for choosing this layer is not detailed. A more principled explanation or heuristic for selecting the optimal layer(s) to apply scaling would make the method even more robust and easier to adopt. The current approach feels slightly post-hoc.

The method involves scaling a single chosen layer's hidden states. A deeper analysis on why a particular layer is optimal or an exploration of scaling multiple layers could provide further insights into the model's internal workings.

**Questions:**

please see the weaknesses.

---

> ### Author Response · Authors · 2025-11-21
> **To Reviewer Fgst**
>
> Thank you for your thoughtful review. Our responses to your questions are provided below.
>
> ---
>
> ### **\[W1\] There is a lack of clear criteria for selecting the optimal layer for scaling, and more analysis is needed.**
>
> The layer selection was determined through hyperparameter tuning, and for a given model, we kept the same configuration even when the benchmark varied. Depending on the model, either a single layer or multiple consecutive layers were selected. For example, LLaVA-OneVision-0.5B required only the first layer, whereas Qwen2.5-VL-3B achieved the best performance when layers 0 through 3 were used.
>
> We primarily selected early consecutive layers, motivated by prior work \[1\] showing that early layers play a key role in visual input encoding (e.g., grounding). In addition, because scaling applied at early layers naturally propagates through subsequent layers, we found that adjusting the early layers provides the most effective strategy in practice.
>
> We also empirically validated the effect of different layer choices. Using the Mantis Benchmark with the Qwen2.5-VL-3B model, which consists of 36 layers (indexed 0–35), we compared applying scaling to layers 0, 1, 2, and 3 individually. The results show that selecting early layers yields the highest performance.
>
> These analyses have been included in the revised manuscript, specifically in Appendix Section A.4 Ablation Study under the paragraph on *Layer Selection*.
>
> | Select Layer | Accuracy |
> | :---- | :---- |
> | Basline | 0.5991 |
> | **0,1,2,3** | **0.6313** |
> | 10,11,12,13 | 0.5991 |
> | 22,23,24,25 | 0.5484 |
> | 32,33,34,35 | 0.5945 |
>
> >\[1\] Yu, Zhuoran, and Yong Jae Lee. "How multimodal llms solve image tasks: A lens on visual grounding, task reasoning, and answer decoding." arXiv preprint arXiv:2508.20279 (2025).

---

### Official Review · Reviewer_4BeG · 2025-10-29

**Soundness:** 3
**Presentation:** 3
**Contribution:** 3
**Rating:** 6
**Confidence:** 3

**Summary:**

This paper addresses a critical limitation of LVLMs, i.e., cross-image information leakage in multi-image tasks, by scaling delimiter token hidden states, a simple yet effective training/inference-cost-free method. Experiments across multi-image (Mantis, MuirBench) and text-only (MultiNews, TQABench) benchmarks show consistent gains, validating its robustness and generality. The analysis of delimiter tokens’ two key properties also provides valuable insights into LVLM attention mechanisms.

**Strengths:**

- Solves cross-image leakage in LVLMs via delimiter scaling, with gains in multi-image/text tasks, no extra cost.
- Analyzes delimiter tokens’ key properties, offering clear theoretical basis for the method.
- Generalizes to text multi-instance tasks, works across models (0.5B–32B), and fits optimized kernels.

**Weaknesses:**

- Though claiming minimal impact on text-image interaction, it only mentions a 10% drop in text-to-image attention scores without detailing how this drop affects downstream cross-modal tasks (e.g., image-text retrieval), leaving uncertainty about real-world cross-modal performance

**Questions:**

- Your work shares similarities with parallel context encoding [1] and attention sink, both focusing on context separation/attention regulation. How does your delimiter scaling method relate to attention entropy (a core factor in Zhang et al.’s work)? Does scaling delimiter states reduce cross-image attention entropy, and if so, how?
- Is there a run-time analysis to show the exact additional inference cost?


[1] Attention Entropy is a Key Factor: An Analysis of Parallel Context Encoding with Full-attention-based Pre-trained Language Models
, Z Zhang et al, ACL 2025.

---

> ### Author Response · Authors · 2025-11-21
> **To Reviewer 4BeG (1/2)**
>
> We appreciate your insightful review. Please find our answers to your questions below.
>
> ---
>
> ### **\[W1\] It is unclear how the 10% drop in text-to-image attention affects downstream cross-modal tasks, leaving uncertainty about real-world cross-modal performance.**
>
> To examine how the decrease in text-to-image attention affects downstream cross-modal performance, we evaluated our method on tasks where understanding the relationship between images and text is essential. MuirBench includes an Image–Text Matching category, which directly measures whether the model can correctly interpret a text snippet and match it to the corresponding visual content. Because this task explicitly tests image–text relational understanding, it serves as a highly relevant downstream cross-modal benchmark. On Qwen2.5-VL-3B, our method improves accuracy in this category from 0.45 to 0.50, surpassing the baseline and indicating that cross-modal understanding remains intact.
>
> We further evaluated our method in a downstream few-shot cross-modal setting. We reorganized a single-image dataset into a 4-shot interleaved format, where each image is sequentially followed by its corresponding question and answer, allowing the model to observe four (image, question, answer) examples before answering the final query. Since this task relies on both image–text interaction and example-based reasoning, it represents a realistic cross-modal scenario. The results show consistent improvements, demonstrating that our method remains effective even when deeper image–text interaction is required.
>
> Specifically, we evaluated the lite versions of OKVQA \[1\] and Vizwiz\_vqa \[2\], curated subsets provided by LMMs-Eval (Huggingface — lmms-lab/LMMs-Eval-Lite). For both datasets, 10% of the samples were used as a validation split for hyperparameter tuning. As shown in the table below, our method consistently improves performance across Qwen2.5-VL-3B, Qwen2.5-VL-7B, and InternVL3-8B.
>
> Taken together, these findings demonstrate that our method preserves text–image interaction and remains effective on downstream cross-modal tasks. The corresponding explanation has been added to the revised manuscript in Section 6.2 Results, under the paragraph titled *Few-Shot Evaluation with Interleaved Examples.*
>
> | Dataset | Model | Qwen2.5-VL-3B | Qwen2.5-VL-7B | InternVL3-8B |
> | :---- | :---- | :---- | :---- | :---- |
> | OKVQA | Baseline | 18.04 | 27.56 | 46.84 |
> |  | \+Ours | **20.00** | **28.24** | **48.68** |
> | Vizwiz | Baseline | 42.38 | 53.7 | 47.04 |
> |  | \+Ours | **42.88** | **54.36** | **50.92** |
>
> >\[1\] Marino, Kenneth, et al. "Ok-vqa: A visual question answering benchmark requiring external knowledge." Proceedings of the IEEE/CVF conference on computer vision and pattern recognition. 2019\.
>
> >\[2\] Gurari, Danna, et al. "Vizwiz grand challenge: Answering visual questions from blind people." Proceedings of the IEEE/CVF conference on computer vision and pattern recognition. 2018\.
>
> ### **\[Q1\] How does your delimiter scaling method relate to attention entropy? Does scaling delimiter states reduce cross-image attention entropy, and if so, how?**
>
> ##### **Difference from Parallel Context Encoding**
>
> We summarize the connection and distinction between our setting and prior work on parallel context encoding as follows.
>
> ###### ***Similarities:***
>
> Both studies address a common research question: how to regulate interactions among multiple contexts within a single input sequence.
>
> ###### ***Differences:***
>
> First, the prior work focuses on LLMs, whereas our analysis centers on LVLMs. As a result, the patterns the model encounters during training and the behaviors observed at inference time differ substantially. Specifically, the prior work reports that, in parallel context encoding, abnormally amplified hidden states of sink tokens cause irregular increases in attention entropy. It also argues that the model encounters a multi-sink pattern at inference time that it has never experienced during training, which ultimately leads to performance degradation.
>
> In contrast, as shown in our analysis and in Figure 1, LVLMs naturally experience multi-sink–like patterns during training due to the structural role of delimiter tokens. Thus, unlike LLMs, LVLMs do not face a new, unseen pattern at inference time, and attention entropy changes are less likely to directly result in performance degradation. For this reason, the phenomena observed in LLMs do not directly transfer to LVLMs. A key contribution of our work is analyzing these behaviors specifically in the LVLM setting.
>
> We have included this discussion in the revised manuscript, specifically in Appendix A.10 *Additional Discussion*.
>
> The response continues in the next comment.

---

> > ### Author Response · Authors · 2025-11-21
> > **To Reviewer 4BeG (2/2)**
> >
> > This comment continues from the previous \[Q1\] response.
> >
> > ##### **Attention Entropy Analysis**
> >
> > We also conducted an attention entropy analysis to further examine how our method affects cross-image attention. We first measured token-level attention entropy in the baseline setting with multi-image inputs and then compared how the entropy changes when our method is applied. The corresponding results are provided in Figure A5 of the revised manuscript.
> >
> > As shown in Figure A5, the baseline exhibits a sharp drop in entropy at the delimiter position between images. This behavior matches the attention patterns in Figure 1(a). For example, immediately before the delimiter of Image 3 (i.e., at the final Query position of Image 2), strong intra-image interaction leads to higher entropy. Once the delimiter token of Image 3 becomes the Query token, however, attention to Images 1 and 2 largely disappears and concentrates on the sink token and the delimiter itself, resulting in lower entropy. In short, the delimiter token absorbs much of the attention and suppresses backward cross-image attention, reducing entropy.
> >
> > At the same time, as highlighted by the red box in Figure 1(a), delimiter tokens in the baseline still allow a non-negligible amount of cross-image attention. In contrast, Figure 4 shows that our method substantially reduces these cross-image interactions. Accordingly, we expected the entropy in the image region to decrease when applying our method. This is confirmed by the orange curve in Figure A5: entropy in the image region is consistently lower than in the baseline, whereas entropy in the text region remains nearly unchanged, indicating that text–image interactions are preserved.
> >
> > These findings are consistent with our earlier analysis: our method suppresses cross-image information leakage while maintaining text–image interaction, and this effect is clearly reflected in the entropy patterns. The full analysis is included in Section A.6, *Analysis of attention entropy changes*, in the revised manuscript.
> >
> > #####
> >
> > ##### **Extend with Entropy mechanisms**
> >
> > We observed that the entropy behavior of delimiter tokens could be leveraged to further extend our method, and we incorporated an additional mechanism based on attention entropy. Specifically, for each delimiter token, we computed the attention entropy over other image tokens and dynamically adjusted the scaling parameter accordingly. When the entropy was higher—indicating a more dispersed attention distribution—we applied stronger scaling.
> >
> > This extended mechanism yields additional performance improvements on the Mantis Benchmark with InternVL3-14B and llava-onevision-0.5B. However, it requires full access to attention maps and is therefore incompatible with FlashAttention, resulting in more than a 2× slowdown in inference. Memory consumption also increases substantially (≈1.6× on average, \+36GB additional, \+54GB peak), exceeding the capacity of typical GPU environments and making the approach practically challenging to deploy.
> >
> > While our main method intentionally prioritizes simplicity and practicality with minimal computational overhead, there may be scenarios where additional performance gains justify higher computational cost. For this reason, we describe this extended mechanism in Appendix Section A.7, including its implementation details, empirical results, and a discussion of its advantages and limitations. We appreciate the reviewer’s suggestion, which allowed us to broaden the scope of the manuscript.
> >
> > | Method | Accuracy |
> > | :---- | :---- |
> > | Baseline (InternVL3-14B) | 71.89 |
> > | Ours | 72.81 |
> > | Ours+Entropy | **73.27** |
> >
> >
> >
> > | Method | Accuracy |
> > | :---- | :---- |
> > | Baseline (llava-onevision-0.5B) | 40.09  |
> > | Ours | 41.01 |
> > | Ours+Entropy | **41.47** |
> >
> >
> >
> > | Method | Time (per sample) | Peak Memory | Avg Memory |
> > | :---- | :---- | :---- | :---- |
> > | Baseline | 0.77s | 60.74GB | 60.04GB |
> > | Ours | 0.77s | 60.74GB | 60.04GB |
> > | Ours+Entropy | 1.78s | 114.76GB | 95.71GB |
> >
> > ### **\[Q2\] Is there a run-time analysis to show the exact additional inference cost?**
> >
> > We empirically verified that our method introduces no additional inference cost. Using four A5000 GPUs, we repeated the experiment three times and report the averaged results. Both the average and peak VRAM consumption were identical to the Baseline, and the inference time also remained unchanged. These results confirm that our method does not incur any additional inference overhead in practice.
> >
> > We have added this analysis to the revised manuscript, specifically in Appendix Section A.5 *Inference Cost*.
> >
> > |  | Avg Memory | Peak Memory | Inference Time |
> > | :---- | :---- | :---- | :---- |
> > | Baseline | 8.27 ± 0 GB | 10.18 ± 0 GB | 100 ± 1s |
> > | Ours | 8.27 ± 0 GB | 10.18 ± 0 GB | 99.67 ± 0.58 s |

---

### Official Review · Reviewer_pZZk · 2025-10-29

**Soundness:** 3
**Presentation:** 2
**Contribution:** 2
**Rating:** 4
**Confidence:** 4

**Summary:**

This paper presents a training-free method for limiting cross-image interactions in LVLMs by scaling the hidden states of delimiter tokens across multiple images. Comprehensive evaluation results on multi-image, multi-document, and multi-table benchmarks are provided to demonstrate the effectiveness of the proposed approach. Both qualitative and quantitative analyses are included to illustrate the role and impact of image delimiter tokens.

**Strengths:**

1. The problem of cross-image information leakage in multi-image LVLM settings is important and worth investigating.
2. The prior analysis on delimiter tokens and the characterization of their key properties is clear and insightful, helping readers better understand the mechanism.
3. The experiments are extensive, covering multiple LVLM families and sizes, four multi-image benchmarks, two multi-document benchmarks, and one multi-table benchmark.

**Weaknesses:**

1. The concept of “sink tokens” has been studied in prior works, and there are also existing methods addressing cross-image leakage. Thus, the novelty and significance of the current findings appear limited.
2. The technical contribution of the proposed method is relatively weak. Scaling the hidden states of delimiter tokens provides only marginal performance gains. For instance, when applied to larger LVLMs such as InternVL3-14B or Qwen2.5-VL-32B, the improvements are minimal (e.g., 42.42 → 42.58).

**Questions:**

1. Could the authors propose a more technically substantial method or extend the current approach with additional mechanisms? Further ablation studies building on the discovered delimiter token properties may help strengthen the contribution and highlight the novelty of the work.
2. The writing could be further polished for clarity, and several figures and tables could be improved for readability. Enhancing the presentation quality would significantly improve the paper.

---

> ### Author Response · Authors · 2025-11-21
> **To Reviewer pZZk (1/4)**
>
> Thank you for your constructive and thoughtful review. We address your questions in the responses below.
>
> ---
>
> ### **\[W1\] The concept of “sink tokens” has been studied in prior works, and there are also existing methods addressing cross-image leakage. Thus, the novelty and significance of the current findings appear limited.**
>
> ##### **Difference from sink token studies**
>
> While prior work has discussed sink tokens extensively, these studies have focused almost exclusively on the \<BOS\> token. In contrast, our work provides the first analysis and detailed characterization of *Image Delimiter Tokens*, a component that has not been examined in earlier research. This offers a novel perspective beyond existing sink-token studies, which treat the phenomenon in a more general context.
>
> ##### **Difference from methods addressing cross-image leakage**
>
> We additionally compared our method with Focus \[1\], a technique specifically designed to mitigate cross-image information leakage. For fairness, we conducted a grid search over 81 hyperparameter configurations and used the best-performing setting for comparison.
>
> Our method consistently outperforms Focus on the Mantis Benchmark. Moreover, Focus requires substantially more memory: for Qwen2.5-VL-7B and InternVL3-8B, it resulted in out-of-memory failures in environments where both the original model and our method run without issues (failing on 24GB and 48GB GPUs, respectively). In terms of both average and peak VRAM usage, our approach is significantly more efficient—using roughly half the peak memory of Focus—and it also achieves faster runtime. These results show that our method provides meaningful performance gains while offering superior resource efficiency.
>
> We have included this content in the revised manuscript, specifically in Section 6.2 Results under the paragraph *Comparison with Focus*.
>
> | Mantis | Qwen2.5-VL-3B | Qwen2.5-VL-7B | InternVL3-1B | InternVL3-8B |
> | :---- | :---- | :---- | :---- | :---- |
> | Baseline | 59.91 | 68.66 | 47.00 | 67.28 |
> | Focus | 58.53 | OOM | 47.93 | OOM |
> | **Ours** | **63.13** | **69.12** | **49.77** | **69.12** |
>
> |  | Memory avg | Memory peak | Time |
> | :---- | :---- | :---- | :---- |
> | Focus | 10.76GB | 21.86GB | 5m 21s |
> | Ours | 8.27GB | 10.18GB | 1m 41s |
>
> >\[1\] Park, Yeji, et al. "Mitigating Cross-Image Information Leakage in LVLMs for Multi-Image Tasks." arXiv preprint arXiv:2508.13744 (2025).
>
> ### **\[W2\] The technical contribution of the proposed method is relatively weak. For instance, when applied to larger LVLMs, the improvement remains minimal.**
>
> ##### **Performance on Larger-Scale Models**
>
> In response to the concern that larger models showed smaller performance gains, we conducted additional experiments on even larger LVLMs. Specifically, we evaluated our method on the Mantis benchmark using Qwen2.5-VL-72B and InternVL3-78B, the largest models in their respective series. Both models showed performance improvements with our method, demonstrating that its effectiveness extends to large-scale models as well. We have added these results to the revised manuscript in Section 6.2 Results, under *Performance on Larger-Scale Models*.
>
> | Mantis | Qwen2.5-VL-72B | InternVL3-78B |
> | :---- | :---- | :---- |
> | Baseline | 74.19 | 74.65 |
> | \+Ours | **75.58** | **76.50** |

---

> > ### Author Response · Authors · 2025-11-21
> > **To Reviewer pZZk (2/4)**
> >
> > ### **\[Q1\] A more technically substantial method or extensions with additional mechanisms would be beneficial, and further ablation studies building on the delimiter token properties.**
> >
> > ##### **Extend the current approach with additional mechanisms**
> >
> > We adopted hidden-state scaling because it is the simplest and most efficient option that remains fully compatible with FlashAttention. Still, as the reviewer suggested, we explored a more substantial extension inspired by prior work \[1\]. We first observed that delimiter tokens show lower attention entropy when they effectively suppress cross-image leakage (Figure A5). Based on this, we implemented an entropy-based variant that dynamically adjusts the scaling factor according to each delimiter token’s attention-weight entropy. We measured the entropy of each delimiter token’s attention distribution over other image tokens and applied stronger scaling when the entropy was higher—that is, when the attention was more dispersed.
> >
> > This extension provides additional performance improvements on the Mantis Benchmark with InternVL3-14B and llava-onevision-0.5B. However, it requires full access to attention maps, making it incompatible with FlashAttention and causing a \>2× slowdown. Memory usage also increases sharply (≈1.6× on average, \+36GB additional, \+54GB peak), exceeding the capacity of common GPUs and making the approach impractical.
> >
> > Thus, while more complex mechanisms can yield extra gains, their computational and memory costs are prohibitive. This highlights the advantage of our original method as a simple and practically deployable solution.
> >
> > Details have been added to Appendix Section A.7.
> >
> > | Method | InternVL3-14B | llava-onevision-0.5B |
> > | :---- | :---- | :---- |
> > | Baseline  | 71.89 | 40.09  |
> > | Ours | 72.81 | 41.01 |
> > | Ours+Entropy | **73.27** | **41.47** |
> >
> >
> >
> >
> >
> > | Method | Time (per sample) | Peak Memory | Avg Memory |
> > | :---- | :---- | :---- | :---- |
> > | Baseline | 0.77s | 60.74GB | 60.04GB |
> > | Ours | 0.77s | 60.74GB | 60.04GB |
> > | Ours+Entropy | 1.78s | 114.76GB | 95.71GB |
> >
> > The response continues in the next comment.
> >
> > >\[1\] Attention Entropy is a Key Factor: An Analysis of Parallel Context Encoding with Full-attention-based Pre-trained Language Models , Z Zhang et al, ACL 2025\.

---

> ### Author Response · Authors · 2025-11-21
> **To Reviewer pZZk (3/4)**
>
> This comment continues from the previous \[Q1\] response.
>
> ##### **Further ablation studies building on the discovered delimiter token properties**
> We conducted the following three further ablation studies.
>
> 1. Ablation study comparing scaling applied individually to Q, K, and V
>
> Because scaling the hidden state affects Q, K, and V through subsequent projections, we conducted an ablation study to isolate the effect on each component. For each of Q, K, and V, we applied scaling after its projection layer. All three variants outperform the Baseline, with K-only scaling yielding the largest improvement. This is likely because amplifying the Key of the delimiter token strengthens its match with Queries, producing a more distinct block-wise attention pattern across image chunks. However, although K-only scaling achieves the strongest single-component result (62.67), it still falls short of the performance of our full method (Ours), which jointly scales Q, K, and V (63.13), suggesting that the effect of K-only scaling on image tagging remains limited. These results indicate that combining Q, K, and V scaling leverages the complementary contributions of all three components, making the joint approach more effective for multi-image understanding. Note that this ablation was conducted using Qwen2.5-VL-3B on the Mantis Benchmark.
>
> | Method | Accuracy |
> | :---- | :---- |
> | Baseline | 59.91 |
> | Q scaling | 61.75 |
> | K Scaling | 62.67 |
> | V Scaling | 61.29 |
> | **Ours** | **63.13** |
>
>
>
> 2. Scaling other tokens
>
> We observed that delimiter tokens inherently help separate images, and our method aims to amplify this behavior by scaling them. To confirm that the improvement indeed comes from the delimiter token, we conducted two ablations on Qwen2.5-VL-3B using the Mantis Benchmark: (1) scaling the first token—which is known to receive strong attention—while leaving the delimiter tokens unchanged, and (2) replacing delimiter tokens with another special token and applying the same scaling.
>
> Scaling the first token yields a small improvement over the Baseline but remains far below Ours. This modest gain likely occurs because the first token acts as a sink token; scaling it amplifies its dominant attention and allows it to absorb some cross-image interactions. However, because it does not provide any image-level tagging function, its effectiveness is limited.
>
> When we replaced delimiter tokens with \<|im\_start|\> and applied scaling to those substitutes, the performance dropped below the Baseline. This shows that simply placing an arbitrary token between images and scaling it does not produce any benefit; only the actual delimiter token leads to improvement.
>
> Together, these results indicate that performance does not improve by scaling an arbitrary token. Instead, the effectiveness of our method arises specifically from the structural and functional role that the delimiter token plays in multimodal models.
>
> |  | Accuracy |
> | :---- | :---- |
> | Baseline | 59.91 |
> | First token scaling | 60.37 |
> | **Ours** | **63.13** |
>
> |  | Accuracy |
> | :---- | :---- |
> | Baseline | 59.91 |
> | Scaled \<im\_start\> as Delim token | 53.00 |
> | **Ours** | **63.13** |
>
> 3. Scaling with a value smaller than 1
>
> Our method scales the delimiter token using factors greater than 1\. To analyze the effect of weakening the delimiter, we additionally evaluated a setting where the scaling factor was set to a value smaller than 1\. With scaling \< 1, the delimiter token is expected to receive insufficient attention, making it difficult to suppress cross-image information leakage. This experiment was performed on the Mantis Benchmark using the Qwen2.5-VL-3B model.
>
> The results confirm this expectation: as shown in Figure A3, performance drops below the Baseline, and as shown in Figure A4, attention map analysis shows that cross-image leakage persists and in some regions becomes even more pronounced. We attribute this degradation to the reduced scaling factor, which prevents the delimiter token from receiving sufficiently strong attention.
>
> We have added these findings to the revised manuscript in Appendix Section A.4, *Ablation Study*.

---

> > ### Author Response · Authors · 2025-11-21
> > **To Reviewer pZZk (4/4)**
> >
> > ### **\[Q2\] The writing could be further polished for clarity, including improvements to several figures and tables.**
> >
> > We revised the manuscript to improve clarity, and the updated content is now reflected in the revised version. All modifications are highlighted in red. For updated figures or tables, we highlighted the captions in red to clearly indicate the changes. For example, Figures 2 and 5 have been updated, and an axis break has been added to their histograms to improve readability. Several tables (e.g., Tables 1, 2, and 3\) have also been adjusted for better presentation.
> >
> > We additionally refined the writing for clarity. For instance, the following sentence was revised as shown below:
> >
> > *Before*: “Figure 5a shows that interactions within the same image remain largely unaffected, indicating that intra-image interaction is preserved.”
> >
> > → *After*: “As seen in the rightmost plot of Figure 5a, the interactions within Image 3 remain largely unaffected, indicating that intra-image interaction is preserved.”
> >
> > *Before*: “Our attention score analysis reveals that while these tokens do contribute to distinguishing images to some extent, cross-image interaction persists.”
> >
> > → *After*: “Our analysis of attention scores shows that although these tokens help distinguish images to some extent, cross-image interaction persists.”
> >
> > *Before*: “To better understand this behavior, we investigate how delimiter tokens help the model distinguish between images and identify two key properties: their ability to absorb attention from other image tokens and their role in reinforcing intra-image interaction.”
> >
> > → *After*: “To better understand this behavior, we examine how delimiter tokens contribute to image separation and identify two key properties: their ability to absorb attention from other image tokens and their role in reinforcing intra-image interaction.”
> >
> > *Before*: “Notably, the fact that M-RoPE, originally developed for temporal distinction in video tasks, contributes to improved performance in multi-image settings further supports our hypothesis that insufficient image distinction is a key factor behind performance drops.”
> >
> > → *After*: “Notably, although M-RoPE was originally designed for temporal distinction in video tasks, it also improves performance in multi-image settings. This further supports our hypothesis that insufficient image distinction is a key cause of performance drops.”

---

### Official Review · Reviewer_exND · 2025-11-03

**Soundness:** 3
**Presentation:** 3
**Contribution:** 3
**Rating:** 6
**Confidence:** 4

**Summary:**

This work focuses on interleaved inputs for multi-modal large language models.

It is motivated by the hypothesis that **adjacent frames carry stronger contextual relevance.**

Unlike previous interleaved approaches that rely on special textual tokens, this work introduces a delimiter token reweighting mechanism.

**Strengths:**

The interleaved image–text processing still lacks clarity in terms of how information flows across modalities. The explanation within MLLMs remains underexplored, though this work makes a valuable attempt to highlight the importance of key regions.

The motivation is clear.

The core idea is **very simple yet interesting**, and the method section is clearly presented. The method do not bring additional training cost and just reweight the hidden states.

**Weaknesses:**

1. The main concern is the **limited scope of evaluation**. The paper focuses primarily on math and multi-view benchmarks, whereas multi-image input represents a special case of __interleaved data__ that can be applied to a broader range of scenarios. The performance under few-shot settings, where multiple instances are concatenated together, remains unclear and differs from the explored benchmarks.

2. The performance improvements are sometimes marginal, suggesting limited generalization.

3. The proposed reweighting operation introduces inconvenience during inference, as it requires modification of hidden states and is therefore applicable only to open-source models.

**Questions:**

1. In Appendix A.1, several details require clarification. For instance, what are the four images and the corresponding text shown? How many samples were used to construct this figure? If the images share visual similarity, would the corresponding results change?

2. Can the proposed method be extended to interleaved inputs (e.g., alternating image–text sequences)?

3. For multi-image inputs, could there be potential attention sink issues similar to those discussed in “When Attention Sink Emerges in Language Models: An Empirical View”?

I may reconsider my evaluation after reading the authors’ rebuttal.

---

> ### Author Response · Authors · 2025-11-21
> **To Reviewer exND (1/2)**
>
> Thank you for your thoughtful review. We have provided our responses to your questions below.
>
> ---
>
> ### **\[W1\] The evaluation scope is limited, and the performance on few-shot settings remains unclear.**
>
> To address this limitation, we conducted additional experiments in the few-shot setting.
>
> We reorganized the single-image dataset into a few-shot setting by constructing 4-shot interleaved inputs, where each image is sequentially followed by its corresponding question and answer. This configuration allows the model to observe four (image, question, answer) examples before generating the final response.
>
> For evaluation, we used the lite versions of OKVQA \[1\] and Vizwiz\_vqa \[2\] provided by LMMs-Eval (Huggingface — lmms-lab/LMMs-Eval-Lite). For both datasets, 10% of the samples were allocated as a validation split for hyperparameter tuning, and the results show consistent improvements across Qwen2.5-VL-3B, Qwen2.5-VL-7B, and InternVL3-8B.
>
> These findings confirm that our method is effective in few-shot settings and generalizes beyond the original data format. The corresponding explanation has been added to the revised manuscript in Section 6.2 Results, under the paragraph titled *Few-Shot Evaluation with Interleaved Examples.*
>
> | Dataset | Model | Qwen2.5-VL-3B | Qwen2.5-VL-7B | InternVL3-8B |
> | :---- | :---- | :---- | :---- | :---- |
> | OKVQA | Baseline | 18.04 | 27.56 | 46.84 |
> |  | \+ Ours | **20.00** | **28.24** | **48.68** |
> | Viswiz | Baseline | 42.38 | 53.7 | 47.04 |
> |  | \+ Ours | **42.88** | **54.36** | **50.92** |
> > \[1\] Marino, Kenneth, et al. "Ok-vqa: A visual question answering benchmark requiring external knowledge." Proceedings of the IEEE/CVF conference on computer vision and pattern recognition. 2019\.
>
> > \[2\] Gurari, Danna, et al. "Vizwiz grand challenge: Answering visual questions from blind people." Proceedings of the IEEE/CVF conference on computer vision and pattern recognition. 2018\.
> ####
>
> ### **\[W2\] The performance improvements are sometimes marginal, suggesting limited generalization.**
>
> To address the reviewer's concern, we conducted additional experiments on larger-scale models to more clearly demonstrate the generalization capability of our method. Specifically, we evaluated our approach on the Mantis benchmark using Qwen2.5-VL-72B and InternVL3-78B, the largest models in their respective series. Both models showed consistent performance improvements with our method, indicating that its effectiveness extends to large-scale models as well. We have added these results to the revised manuscript in Section 6.2, under Performance on *Larger-Scale Models*.
>
> | Mantis | Qwen2.5-VL-72B | InternVL3-78B |
> | :---- | :---- | :---- |
> | Baseline | 74.19 | 74.65 |
> | \+ Ours | **75.58** | **76.50** |
>
> We also conducted additional comparative experiments to better illustrate the extent of the performance improvements. For this purpose, we compared our method with Focus \[1\], a technique specifically designed to mitigate cross-image information leakage, and performed a grid search over 81 hyperparameter configurations to ensure a fair comparison, using the best-performing setting for evaluation.
>
> The experimental results show that our method consistently outperforms Focus across the Mantis Benchmark. Focus also requires substantially more memory, causing out-of-memory errors for models such as Qwen2.5-VL-7B and InternVL3-8B, even in environments where both the original model and our method run without issues. In terms of VRAM usage, our approach is significantly more efficient, with peak memory consumption being roughly half that of Focus, and it also achieves faster runtime.
>
> These findings demonstrate that our method provides meaningful performance gains while maintaining superior resource efficiency, rather than yielding merely marginal improvements. This content has been incorporated into the revised manuscript, specifically in Section 6.2 Results under the paragraph titled *Comparison with Focus*.
>
>
> | Mantis | Qwen2.5-VL-3B | Qwen2.5-VL-7B | InternVL3-1B | InternVL3-8B |
> | :---- | :---- | :---- | :---- | :---- |
> | Baseline | 59.91 | 68.66 | 47.00 | 67.28 |
> | Focus | 58.53 | OOM | 47.93 | OOM |
> | **Ours** | **63.13** | **69.12** | **49.77** | **69.12** |
>
> |  | Memory avg | Memory peak | Time |
> | :---- | :---- | :---- | :---- |
> | Focus | 10.76GB | 21.86GB | 5m 21s |
> | Ours | 8.27GB | 10.18GB | 1m 41s |
>
> >\[1\] Park, Yeji, et al. "Mitigating Cross-Image Information Leakage in LVLMs for Multi-Image Tasks." arXiv preprint arXiv:2508.13744 (2025).

---

> > ### Author Response · Authors · 2025-11-21
> > **To Reviewer exND (2/2)**
> >
> > ### **\[W3\] The proposed method is applicable only to open-source models.**
> >
> > The fact that our method is currently demonstrated only on open-source models is not a limitation specific to our approach; it is a general constraint shared by existing techniques that rely on internal representations such as hidden states or attention. While external users of proprietary systems cannot access these internal signals, model developers can, and integrating our lightweight reweighting mechanism into commercial models would pose little technical difficulty. Therefore, our method should not be viewed as inherently limited to open-source models but rather as an intuitive and easily adoptable improvement for commercial systems as well.
> >
> > We have added this clarification to the revised manuscript, specifically in Appendix Section A.9, *Additional Limitation*.
> >
> > ### **\[Q1\] Appendix A.1 requires further clarification, and requires clarification on whether the results change when the images are visually similar.**
> >
> > To clarify the details in Appendix A.1, which further analyzes the role of image-delimiter tokens, we added a figure in the revised manuscript (Figure A6) that includes the four images and their corresponding text. Although this figure is based on a single example, we observed the same pattern across the vast majority of samples. Additional qualitative examples have been included in Appendix Section A.8 (*Additional Qualitative Results*). Furthermore, the quantitative experiment in Table A1 is conducted on the full Mantis dataset, showing that the importance of the delimiter token consistently appears across a wide range of samples.
> >
> > The results remain stable even when the images are visually similar. We examined cases where multiple images share nearly identical backgrounds and differ only in facial expressions. As shown in Figure A7, the delimiter token still produces a clear separation, forming a triangular boundary pattern in the attention map. In contrast, when the delimiter token is removed or replaced with another special token (e.g., \<|im\_start|\>), the separation between images becomes unclear. This demonstrates that the delimiter token plays a crucial role in distinguishing images even under high visual similarity. Additional supporting figures are included in Appendix Section A.8.
> >
> > ### **\[Q2\] Can the proposed method be extended to interleaved inputs (e.g., alternating image–text sequences)?**
> >
> > Through the few-shot experiments described in \[W1\], we verified that our method extends to interleaved inputs where multiple images and text segments alternate. The method continues to perform consistently well in these settings, confirming its applicability beyond the original image-first input structure.
> >
> > ####
> >
> > ### **\[Q3\] For multi-image inputs, could there be potential attention sink issues similar to those discussed in “When Attention Sink Emerges in Language Models: An Empirical View”?**
> >
> > Attention sink phenomena also appear in multi-image inputs. Similar to LLMs, the first token in LVLMs acts as a sink token, but the pattern differs from what is reported in the referenced LLM work. In multi-image settings, delimiter tokens absorb a substantial amount of attention, which reduces the relative attention assigned to the sink token.
> >
> > This can be observed by comparing Figure A8 (d) and (e) in Appendix. The leftmost vertical column corresponds to the sink token at the beginning of the system prompt. In (e), where no delimiter tokens are present, the sink token receives strong attention, whereas in (d), the delimiter tokens draw attention away, resulting in reduced attention on the sink token. Thus, while sink-token behavior does occur in multi-image inputs, the presence of delimiter tokens redistributes attention and weakens the dominance of the sink token.
> >
> > We already cite the referenced work in the main paper and have added a more detailed discussion under the section “Sink tokens in large language models.”

---

> > > ### Comment · Reviewer_exND · 2025-11-28
> > >
> > > My questions are also addressed especially the attention map explanation.

---

> > ### Comment · Reviewer_exND · 2025-11-28
> >
> > Few-Shot Evaluation with Interleaved Input remains a hot topic and the rebuttal addressed my concern about it. All these complementary experiments should be preserved and released in the future version.

---

> ### Author Response · Authors · 2025-11-28
>
> Thank you for the positive feedback. We have incorporated all the points you mentioned, and they will be reflected in the final version. Your feedback has helped improve the clarity and quality of the submission.

---

### Comment · Area_Chair_TxNK · 2025-11-23
**The authors' rebuttal is available. Please read, comment, and discuss.**

Dear Reviewers,

Thanks for your time and effort in reviewing ICLR2026 submissions. The authors have provided their responses to your review. Please read and raise your further comments, and discuss with the authors.

Best regards,

Your AC

---

### Author Response · Authors · 2025-11-23
**General Response**

We sincerely thank **Reviewers exND, pZZk, 4BeG and Fgst** for providing detailed, thoughtful evaluations and highly valuable feedback. We are grateful for the encouraging remarks highlighted throughout the reviews. We appreciate the positive comments from all reviews:


* **Reviewer exND**
1. Provides a helpful attempt to clarify information flow.
2. Clear motivation with a simple, well-explained method.
3. Requires no training cost and only reweights hidden states.
* **Reviewer pZZk**
1. Addresses cross-image leakage in multi-image LVLMs.
2. Offers clear and insightful analysis of delimiter tokens.
3. Provides extensive experiments across diverse models and benchmarks.
* **Reviewer 4BeG**
1. Mitigates cross-image leakage via delimiter scaling with no inference cost.
2. Gives a clear theoretical basis through delimiter token analysis.
3. Generalizes across sizes and tasks.
* **Reviewer Fgst**
1. Provides clear analysis of delimiter behavior with strong motivation.
2. The method is simple without training or inference overhead.
3. Shows consistent effectiveness across models and domains.

---

We have thoroughly revised the manuscript to address all reviewers’ comments. Newly added or modified content is highlighted in blue in the revised version, while sentences, tables, and figures that were refined for clarity are highlighted in red. Below, we summarize the main changes incorporated in this revision.

* **Reviewer exND**
1. Added few-shot evaluation, larger-model evaluations, and comparison with Focus in Section 6.2. **\[W1, W2, Q2\]**
2. Included the open-source–only limitation in Appendix A.9. **\[W3\]**
3. Provided details in Appendix A.1 and additional qualitative results in A.1 and A.8. **\[Q1\]**
4. Added sink-token explanation for multi-image inputs in Related Work. **\[Q3\]**
* **Reviewer pZZK**
1. Added discussion in Related Work distinguishing sink tokens from our contribution. **\[W1\]**
2. Included larger-model evaluations and comparison with Focus in Section 6.2. **\[W2\]**
3. Added an entropy-based extension in Appendix A.7. **\[Q1\]**
4. Provided further ablation studies (Scaling Q, K, and V, Scaling other tokens, and Scaling with a value smaller than 1\) to Appendix Section A.4. **\[Q1\]**
5. Revised sentences for clarity and highlighted them in red. **\[Q2\]**
* **Reviewer 4BeG**
1. Included few-shot results to assess cross-modal performance in Section 6.2. **\[W1\]**
2. Added attention entropy analysis and an additional mechanism in Appendix A.6 and A.7. **\[Q1\]**
3. Provided the inference cost results in Appendix Section A.5. **\[Q2\]**
* **Reviewer 4BeG**
1. Added layer-selection details and ablations in Appendix A.4. **\[W1\]**

---

Additional responses to the comments are provided in the corresponding individual comments.

---

### Meta-Review · Area_Chair_8sPv · 2026-01-06

**Summary:**

This paper addresses the critical issue of cross-image information leakage in Large Vision-Language Models (LVLMs) when processing multiple images. All concerns have been satisfactorily addressed, and the paper has received consistently positive scores. Accept.

**Reviewer Concerns:**

Reviewer exND

- Provides a helpful attempt to clarify information flow.
- Clear motivation with a simple, well-explained method.
- Requires no training cost and only reweights hidden states.

Reviewer pZZk

- Addresses cross-image leakage in multi-image LVLMs.
- Offers clear and insightful analysis of delimiter tokens.
- Provides extensive experiments across diverse models and benchmarks.

Reviewer 4BeG

- Mitigates cross-image leakage via delimiter scaling with no inference cost.
- Gives a clear theoretical basis through delimiter token analysis.
- Generalizes across sizes and tasks.

Reviewer Fgst

- Provides clear analysis of delimiter behavior with strong motivation.
- The method is simple without training or inference overhead.
- Shows consistent effectiveness across models and domains.

**Reviewer Scores:**

The paper received initial scores of 4, 6, 6, and 6. Post-rebuttal, Reviewer exND (who gave the initial 4) expressed that their concerns were well-addressed.

---

### Decision · Program_Chairs · 2026-01-26

Accept (Poster)